# Diffusion-Guided Graph Data Augmentation

**Maria Marrium[1],    Arif Mahmood[1],    Muhmmad Haris Khan[2],**
**Muhammad Saad Shakeel[3],    Wenxiong Kang[3]**

[1]Information Technology Univeristy, Lahore, Pakistan
[2]MBZUAI, Abu Dhabi, UAE
[3]South China University of Technology, Guangdong, China
{phdcs22002, arif.mahmood}@itu.edu.pk,   muhammad.haris@mbzuai.ac.ae,
{saadshakeel, auwxkang}@scut.edu.cn

## Abstract

Graph Neural Networks (GNNs) have achieved remarkable success in a wide range of applications. However, when trained on limited or low-diversity datasets, GNNs are prone to overfitting and memorization, which impacts their generalization. To address this, graph data augmentation (GDA) has become a crucial task to enhance the performance and generalization of GNNs. Traditional GDA methods employ simple transformations that result in limited performance gains. Although recent diffusion-based augmentation methods offer improved results, they are sparse, task-specific, and constrained by class labels. In this work, we propose a more general and effective diffusion-based GDA framework that is task-agnostic and label-free. For better training stability and reduced computational cost, we employ a graph variational auto-encoder (GVAE) to learn a compact latent graph representation. A diffusion model is used in the learned latent space to generate both consistent and diverse augmentations. For a fixed augmentation budget, our algorithm selects a subset of samples that would benefit the most from the augmentation. To further improve performance, we also perform test-time augmentation, leveraged by the label-free nature of our method. Thanks to the efficient utilization of GVAE and latent diffusion, our algorithm significantly enhances machine learning safety measures, including calibration, robustness to corruptions, and prediction consistency. Moreover, our method has shown improved robustness against four types of adversarial attacks and achieves better generalization performance. To demonstrate the effectiveness of the proposed method, we compare it with 30 existing methods on 12 benchmark datasets across node classification, link prediction, and graph classification in various learning settings, including semi-supervised, supervised, and long-tailed data distributions. Code is available at https://github.com/MariaMarrium/D-GDA.

## 1   Introduction

Graph Neural Networks (GNNs) have achieved remarkable success across diverse domains, such as social networks [29, 16], recommendation systems [77, 57], and molecular property prediction [4, 53]. However, their expressive power makes them prone to overfitting and memorization, particularly when trained on limited, imbalanced, or low-diversity graph datasets [54, 90, 36, 7, 87]. These limitations undermine their generalization to diverse test scenarios, posing a significant challenge for real-world applications [92]. To address this challenge, Graph Data Augmentation (GDA) has emerged as a promising strategy to enhance the generalization of GNNs by improving the diversity of training data  [88, 45]. Traditional GDA methods are based on static transformations, such as edge modifications [7, 17], node feature alterations [31, 84], or subgraph manipulations [2, 72].

39th Conference on Neural Information Processing Systems (NeurIPS 2025).

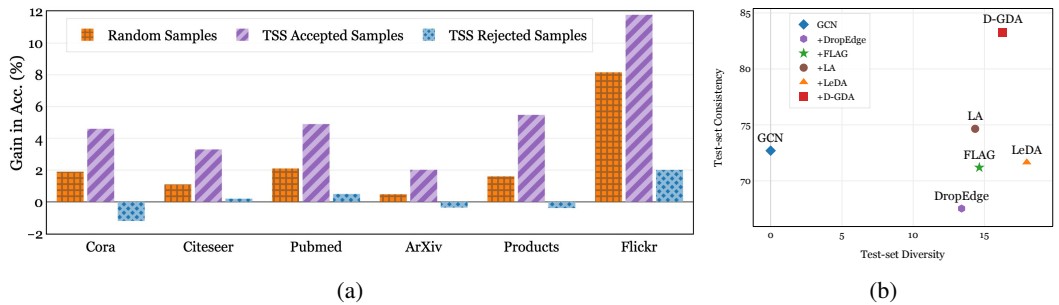

Figure 1: a) Node classification performance comparison: baseline GCN (zero-line), augmenting 20% random, 20% TSS-rejected, and 20% TSS-selected samples. (b) Diversity vs. consistency averaged over the same six datasets. Top-right corner shows a good balance of diversity vs. consistency.

However, these approaches often fail to balance *consistency* (preserving the original labels) and *diversity* (introducing meaningful variations), as illustrated in Figure 1b. Recent diffusion-based GDA methods, such as Data-Centric Transfer (DCT) [40] and Diffusion on Graph (DoG) [71], leverage diffusion models to generate synthetic graphs or nodes, offering improved results. However, these methods are task-specific and rely on class labels that limit their applicability to general tasks. Moreover, the GDA methods have not been thoroughly evaluated for robustness against adversarial attacks and machine learning (ML) safety measures, such as calibration, corruption, and prediction consistency (see Table 1).

To overcome these limitations, we propose **D**iffusion-based **G**raph **D**ata **A**ugmentation (**D-GDA**), which is a novel, applicable to different graph tasks and label-free framework. It improves the generalization of GNNs in various graph tasks, including node classification, link prediction, and graph classification. The D-GDA introduces three key concepts: (1) *Target sample selection*, which identifies candidates for effective augmentation to maximize performance gain. It helps D-GDA focus on challenging regions in the training data space (Figure 1a); (2) *Graph variational auto-encoder*, to encode local graph structure and semantics into a compact latent space to reduce computational cost without affecting performance; (3) *Latent Diffusion Model*, which generates consistent and diverse synthetic nodes and edges conditioned on neighborhood embeddings. It ensures that the generated augmentations preserve local graph structure while introducing meaningful diversity (Figure 1b). Unlike prior diffusion-based GDA methods, the proposed D-GDA is label-free, and therefore, it can be used to augment the test samples to improve performance. The test-time augmentation [27] has not been well studied in the graph data. We observe a consistent performance gain by performing test-time augmentation in graph and node classification tasks.

Table 1: A comparison of SOTA GDA methods: 1) **Basic building blocks:** graph auto-encoder (GAE), Graph Variational auto-encoder (GVAE), and Diffusion models. 2) **Supported tasks:** Node Classification (NC), Link Prediction (LP), and Graph Classification (GC). 3) **Robustness evaluations:** adversarial robustness (Adv) and machine learning safety measures (MSM). 4) **Learning Settings:** semi-supervised, Supervised, and Long-tailed (LT). 5) **Test-Time Augmentation** (TTA).

| Methods | Basic Building Blocks | | | Supported Tasks | | | Robustness | | Learning Setting | | | TTA |
|---|---|---|---|---|---|---|---|---|---|---|---|---|
| | GAE | GVAE | Diff. | NC | LP | GC | Adv | MSM | Semi | Sup. | LT | |
| DropEdge [54] | ✗ | ✗ | ✗ | ✓ | ✗ | ✗ | ✗ | ✗ | ✓ | ✗ | ✗ | ✗ |
| GAug [90] | ✓ | ✗ | ✗ | ✓ | ✗ | ✗ | ✗ | ✗ | ✓ | ✗ | ✗ | ✗ |
| FLAG [31] | ✗ | ✗ | ✗ | ✓ | ✓ | ✓ | ✗ | ✗ | ✓ | ✓ | ✗ | ✗ |
| LA [42] | ✗ | ✓ | ✗ | ✓ | ✓ | ✓ | ✗ | ✗ | ✓ | ✓ | ✗ | ✗ |
| GraphPatcher [25] | ✗ | ✗ | ✗ | ✓ | ✗ | ✗ | ✗ | ✗ | ✓ | ✗ | ✗ | ✓ |
| DCT [40] | ✗ | ✗ | ✓ | ✗ | ✗ | ✓ | ✗ | ✗ | ✗ | ✓ | ✗ | ✗ |
| LeDA [41] | ✗ | ✗ | ✗ | ✓ | ✗ | ✗ | ✓ | ✗ | ✓ | ✗ | ✗ | ✗ |
| DoG [71] | ✓ | ✗ | ✓ | ✓ | ✗ | ✗ | ✗ | ✗ | ✓ | ✗ | ✗ | ✗ |
| GraphVCM [52] | ✗ | ✗ | ✗ | ✓ | ✗ | ✗ | ✗ | ✗ | ✗ | ✗ | ✓ | ✗ |
| D-GDA (ours) | ✗ | ✓ | ✓ | ✓ | ✓ | ✓ | ✓ | ✓ | ✓ | ✓ | ✓ | ✓ |

The proposed D-GDA offers several advantages over existing methods, including performance gains, improvements in ML safety measures, robustness to adversarial attacks, and lower computational cost compared to other diffusion-based methods. ML safety measures include calibration, corruption, and prediction consistency. D-GDA improves robustness to Random, DICE, GF, and Meta-Attacks. D-GDA is evaluated in semi-supervised, supervised, and long-tailed classification settings, demonstrating consistent improvements. D-GDA also achieves a good balance of diversity vs. consistency over existing compared methods. Through extensive experiments on 12 benchmark datasets for node classification, link prediction, and graph classification, D-GDA has shown excellent performance compared to 30 state-of-the-art methods.

**Contributions: (1)** We propose D-GDA, a label-free, diffusion-based graph data augmentation framework that excels in node classification, link prediction, and graph classification across semi-supervised, supervised, and long-tailed settings. It supports test-time augmentation for enhanced performance. **(2)** D-GDA leverages a variational auto-encoder and latent diffusion model for proposed neighborhood-aware node generation. **(3)** D-GDA introduces a Target Sample Selector to select effective candidates resulting in overall performance improvement for a fixed augmentation budget. **(4)** D-GDA enhances ML safety measures including calibration, resistance to corruption, and consistency. It is more robust against adversarial attacks (Random, DICE, GF, Meta-Attack) and converges to a flatter minima for improved generalization and reduced over-smoothing.

## 2  Related Work

**Traditional Graph Data Augmentation** methods do not rely on generative models. Instead, they apply augmentations by modifying the graph structure or features—such as adding/removing edges [7, 90], altering node features [1, 68, 73], or manipulating subgraphs [48, 72, 14, 65, 82]. Some methods also adopt hybrid strategies that combine multiple techniques [80, 64, 83, 32, 76]. For instance, DropEdge [54] randomly removes edges for regularization, and FLAG [31] introduces gradient-based feature perturbations. GAMS [2] partitions graphs into clusters, computes inter-cluster similarity, and swaps the most similar clusters. GeoMix [91] incorporates spatial structure through geometry-aware mixup. Notably, these methods modify existing graphs without increasing dataset size, whereas D-GDA explicitly enlarges the training set by generating new, task-relevant augmented samples. **Generative Graph Data Augmentation** methods leverage generative models to enrich graph data. LA [42] uses a Variational Graph Autoencoder (VGAE) to generate local node features for augmentation. DCT [40] focuses on graph classification, generating entire graphs conditioned on graph-level properties. DoG [71] targets node classification and employs a diffusion model conditioned on class labels to synthesize node features. In addition to these, there are methods that generate molecular graphs conditioned on input graphs using both GVAE [81, 58] and diffusion models [24, 69, 38, 5, 9]. However, these approaches primarily focus on generating valid molecules, and their effectiveness for improving graph classification performance remains unexplored. In contrast, D-GDA is a versatile framework applicable to node classification, link prediction, and graph classification. Rather than conditioning on class labels, it introduces neighborhood-aware conditioning to guide the diffusion model denoising process, enabling context-aware and task-relevant augmentations that enhance performance across diverse graph tasks. **Class Imbalance Handling** methods tackle class imbalance in graph data by primarily augmenting minority class nodes to balance the training set. Recent approaches include GraphSMOTE [87], ImgAGN [51], GraphENS [49], GraphSHA [37], and GraphVCM [52]. These methods typically use mixup to generate new node features and connect them either to the neighbors of both mixed nodes or just the target node. A comprehensive survey of such techniques is provided by [44]. In contrast, D-GDA focuses on difficult-to-learn nodes and employs GVAE and LDM models to generate augmentations with selected nodes. **Masked Autoencoders** learn robust representations by masking parts of the input and training the model to reconstruct the missing content. Originally popularized in NLP with masked language modeling [12], this approach has been adapted for graphs via Masked Graph Modeling (MGM) [47, 34, 66], which predicts masked node features and edges. We leverage MGM into our GVAE framework to enhance the quality of latent representations for our latent diffusion model.

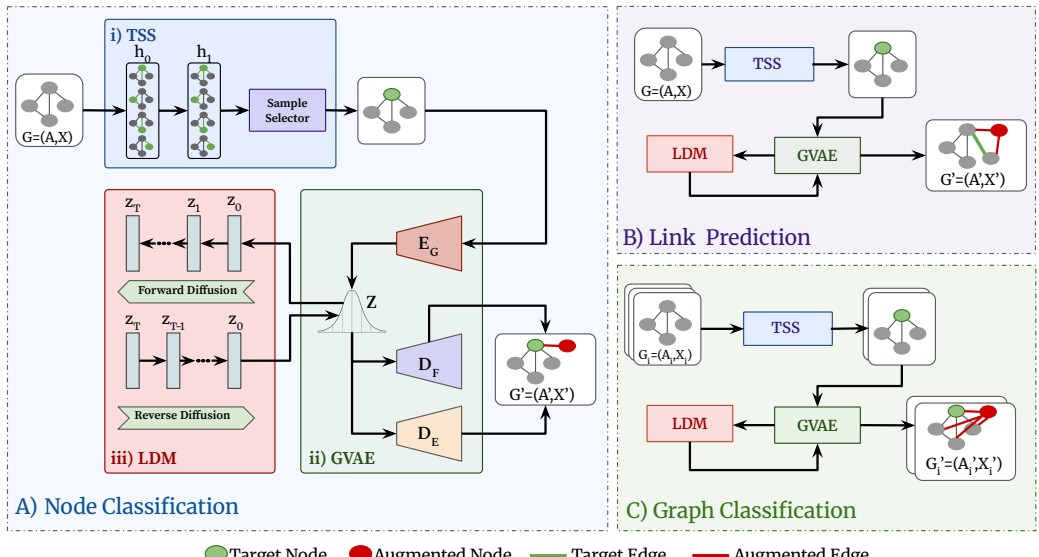

Target Node ● Augmented Node ● Target Edge ── Augmented Edge ──

Figure 2: Overall architecture of the proposed D-GDA framework: A) **Node classification:** i) We propose Target Sample Selector (TSS) to select challenging-to-learn nodes. ii) We harness Graph Variational Autoencoder (GVAE) to encode the graph structure into a latent representation. GVAE decoders reconstruct augmented node features and edges from the latent space. iii) Latent Diffusion Model (LDM) generates a neighborhood-aware latent representation. B) **Link Prediction:** TSS selects challenging-to-learn target links. GVAE encodes graph latent representation. LDM generates augmented node latent representation based on average aggregated edge neighborhood. C) **Graph Classification:** TSS selects challenging-to-learn target graphs. GVAE encodes each graph to latent space. Maximum degree nodes are selected as target nodes. Augmented nodes are generated using LDM based on target node neighborhood.

## 3 Methodology

**Preliminaries:** Let $G = (\mathcal{V}, \mathcal{E})$ be a graph, where $\mathcal{V} = \{v_i\}_{i=1}^{n}$ be the set of $n$ nodes and $\mathcal{E}$ be that set of edges. Each node is associated with a feature vector $x_i \in \mathcal{R}^d$ and $\boldsymbol{X} = [\boldsymbol{x}_1, \boldsymbol{x}_2, \ldots, \boldsymbol{x}_n]^\top \in \mathcal{R}^{n \times d}$ be the feature matrix. The edges are encoded in adjacency matrix $\boldsymbol{A} \in \mathcal{R}^{n \times n}$, where $\boldsymbol{A}_{ij} = w_{ij}$ if there is an edge from $v_i$ to $v_j$, and 0 otherwise. **Graph Neural Networks (GNNs)** learn low-dimensional node representations $\mathbf{H} \in \mathcal{R}^{n \times d'}$ by aggregating neighborhood information. A common GNN variant, Graph Convolutional Network (GCN) [29], updates node embeddings layer-wise as:

$$\mathbf{H}^{(l+1)} = \sigma \left( \tilde{\mathbf{D}}^{-\frac{1}{2}} \tilde{\mathbf{A}} \tilde{\mathbf{D}}^{-\frac{1}{2}} \mathbf{H}^{(l)} \mathbf{W}^{(l)} \right), \quad (1)$$

where, $\tilde{\mathbf{A}} = \mathbf{A} + \mathbf{I}$ is the adjacency matrix with self-loops, $\tilde{\mathbf{D}}$ is its corresponding degree matrix, $\mathbf{H}^{(0)} = \mathbf{X}$ is the initial node feature matrix, $\mathbf{W}^{(l)} \in \mathcal{R}^{d_l \times d_{l+1}}$ is the weight matrix at layer $l$, and $\sigma(\cdot)$ is a non-linear activation function such as ReLU. The final layer output, $\mathbf{H}$, provides node embeddings for downstream tasks. **Node Classification:** For a graph $G$ with $\mathcal{V}_l \subseteq \mathcal{V}$ labeled nodes with labels $\boldsymbol{Y_N} = \{\boldsymbol{y}_i\}_{i=1}^{|V_l|}$, where $\boldsymbol{y}_i \in \mathcal{R}^c$ is a one-hot vector for $c$ classes. The $\boldsymbol{H}$ is passed through a softmax layer for classification. **Link Prediction:** For a graph $G$, node embeddings $\boldsymbol{H}$ are used with a scoring function (e.g., inner product) to predict edge (link) likelihood between node pairs. **Graph Classification:** For a set of graphs $\mathcal{G} = \{G_1, \ldots, G_m\}$ with labels $\boldsymbol{Y_G}$, the node embeddings $\boldsymbol{H}$ are aggregated using a permutation-invariant readout layer, for example, mean pooling, to obtain the graph representation $\boldsymbol{r}_{G_i}$, followed by the classification layer.

### 3.1 Proposed Diffusion-based Graph Data Augmentation (D-GDA) Framework

Our proposed D-GDA framework enhances GNN generalization by generating diverse augmentations. D-GDA comprises of three components: (1) a new Target Sample Selector (TSS) to identify samples

that maximize augmentation advantage, (2) a Graph Variational Autoencoder (GVAE) to learn compact latent representations to reduce computational complexity, and (3) a Latent Diffusion Model (LDM) to generate augmentations which we propose to be conditioned on target node neighborhood. The overall D-GDA framework is shown in Figure 2.

### 3.1.1 Target Sample Selector (TSS)

Given a fixed augmentation budget, our TSS selects a subset of samples that would benefit the most from augmentation, thereby improving overall performance. In Figure 1a, a comparison is shown for node classification using a budget of 20%. A comparison of Randomly-selected, TSS-selected, and TSS-rejected samples demonstrates the importance of this stage. For this purpose, we train a baseline GNN and compute the entropy ($e_s$) of the predicted class probabilities for each sample $s$: $e_s = -\sum_{i=1}^{c} p_s^{(i)} \log p_s^{(i)}$, where $c$ is the number of classes and $p_s^{(i)}$ is the prediction probability for $i$-th class. High entropy indicates prediction uncertainty, marking samples as difficult to learn. The samples are sorted in descending order based on entropy $e_s$. If the augmentation budget corresponds to $k$-samples, the top-$k$ samples are selected as augmentation targets from the sorted list. The bottom $k$ samples of this list are considered as TSS-rejected samples.

### 3.1.2 Graph Variational Autoencoder (GVAE)

In our proposed framework, the latent diffusion model (LDM) is used for the generation of augmented samples. Training of LDM is more stable and computationally efficient in latent representations [21]. Variational graph autoencoders learn expressive latent representations of graph data, capturing both node features and local structure [30, 34, 47]. Therefore, we employ a variant of this architecture for compact latent representation learning, resulting in reduced computational complexity and improved performance. Our GVAE consists of a GCN-based encoder and two decoders for node feature and edge reconstruction. The GVAE consists of a GCN-based encoder and two decoders for node feature and edge reconstruction. **GVAE Encoder** maps node feature matrix $\mathbf{X}$ and the graph adjacency matrix $\mathbf{A}$ to a latent space representation by learning a Gaussian distribution. It first applies $L-1$ layers of graph convolution to compute intermediate node representations $\mathbf{H}^{L-1}$ (See Eq. 1). Then, it is used to parameterize the mean and variance of a Gaussian distribution: $\boldsymbol{\mu} = f_\mu(\mathbf{H}^{L-1}, \mathbf{A}), \quad \log \boldsymbol{\sigma}^2 = f_\sigma(\mathbf{H}^{L-1}, \mathbf{A})$, where, $f_\mu$ and $f_\sigma$ denote separate GCN layers acting as linear transformations. To enable gradient-based learning through stochastic sampling, the latent embedding $\mathbf{Z}$ is drawn using the reparameterization trick [28]: $\mathbf{Z} = \boldsymbol{\mu} + \boldsymbol{\sigma} \odot \boldsymbol{\epsilon}, \quad \boldsymbol{\epsilon} \sim \mathcal{N}(\mathbf{0}, \mathbf{I})$, where $\odot$ denotes element-wise multiplication. The resulting vector $\mathbf{Z} \in \mathcal{R}^{n \times d_z}$ is the latent representation of $G$. **GVAE Feature Decoder** reconstructs the original node features $\mathbf{x}_i$ from the latent variable $\mathbf{z}_i$ using a multilayer perceptron: $\hat{\mathbf{x}}_i = f_{\text{feat}}(\mathbf{z}_i)$. Feature decoder constraints the latent space to preserve semantic information about node attributes and serves as an auxiliary supervision signal. **GVAE Link Predictor** reconstructs the graph structure by estimating the probability of a link (edge) between node pairs $(v_i, v_j)$ based on their latent representation: $\hat{p}_{ij} = \sigma\left(f_{\text{link}}(\mathbf{z}_i \| \mathbf{z}_j)\right)$ where $\sigma(\cdot)$ is the sigmoid function and $\|$ shows concatenation operation. Link predictor constraints the model to learn structural dependencies and reconstruct the adjacency matrix from the latent space $\mathbf{Z}$. **Regularization:** To improve effectiveness of GVAE, we use two different regularization strategies, including edge masking [34] and feature masking [47]. Edge masking randomly removes a subset of edges during training, forcing the model to reconstruct them from partial graph. Feature masking randomly zeros out a fixed percentage of node features coefficient, encouraging generalization. **GVAE Loss Function:** The GVAE is optimized with an overall loss: $\mathcal{L}_{\text{GVAE}} = \mathcal{L}_{\text{edge}} + \lambda_1 \mathcal{L}_{\text{feat}} + \lambda_2 \mathcal{L}_{\text{KL}}$ where, $\mathcal{L}_{\text{feat}} = \sum_i \|\mathbf{x}_i - \hat{\mathbf{x}}_i\|_2^2$ is the feature reconstruction loss that measures the Euclidean distance between original and reconstructed node features, $\mathcal{L}_{\text{edge}} = -\sum_{(i,j) \in E} \log \hat{p}_{ij} - \sum_{(i,j) \notin E} \log(1 - \hat{p}_{ij})$ is the link prediction loss with sub-sampling of non-edges for balancing both type of samples, $\mathcal{L}_{\text{KL}} = D_{\text{KL}}\left[q(\mathbf{z}_i|\mathbf{x}_i) \| \mathcal{N}(0, \mathbf{I})\right]$ is the KL divergence loss used to regularize the learned latent distributions $q(\mathbf{z}_i|\mathbf{x}_i)$ to match a standard Gaussian prior, and $\lambda_1, \lambda_2$ are hyperparameters used as reported by [30].

### 3.1.3 Latent Diffusion Model (LDM)

The LDM generates diverse augmentations in the GVAE latent space by leveraging a Denoising Diffusion Probabilistic Model (DDPM) [21]. This approach is more efficient than diffusion in the input graph space due to improved training stability and reduced augmentation overhead. In addition,

it can effectively capture local structure through conditioning. In the **forward (diffusion) process**, Gaussian noise is gradually added to a clean latent vector $\mathbf{z}_i^0$ over a sequence of $T$ time steps. At each step $t \in \{1, \ldots, T\}$, the noisy latent vector $\mathbf{z}_t$ is computed as: $\mathbf{z}_i^t = \sqrt{\bar{\alpha}_t}\,\mathbf{z}_i^0 + \sqrt{1 - \bar{\alpha}_t}\,\boldsymbol{\epsilon}$, where, $\boldsymbol{\epsilon} \sim \mathcal{N}(0, \mathbf{I})$, $\bar{\alpha}_t = \prod_{s=1}^{t} \alpha_s$, where $\alpha_s \in (0, 1)$ are predefined noise schedule parameters controlling the rate at which noise is injected. This process transforms a structured latent vector into a nearly pure Gaussian noise vector as $t \to T$.

**Reverse Process with Proposed Conditioning:** The existing LDM reverse process aims to recover the clean latent representation $\mathbf{z}_i^0$ solely from a noisy latent vector $\mathbf{z}_i^t$. Such a reverse process may generate augmentations far from the target samples resulting in performance degradation. To handle this issue DoG [71] constrained noise recovery process by class-labels. However, due to intra-class variations it does not ensure the generated augmentation to be close to the target sample. Also, the requirement of class-labels restricts the augmentation to training set only. In contrast, we propose the generation process to be constrained using target-node neighborhood which better encodes a difficult to learn regions. Being label-free, D-GDA is able to extend augmentations beyond training set for better utilization the graph structure, resulting in improved performance. Our *proposed conditioning vector* $\mathbf{c}_i^0$, encodes the local topology by mean aggregating the one-hop neighborhood embeddings $\mathbf{z}_j^0$ of node $v_i$: $\mathbf{c}_i^0 = \mathbb{E}_{j \in \mathcal{N}(v_i)} \mathbf{z}_j$. A denoising network $\epsilon_\theta$, implemented as a 1D-UNet [55], predicts the noise $\boldsymbol{\epsilon}$ using $\mathbf{z}_i^t$, and $\mathbf{c}_i^0$, formulated as $\hat{\boldsymbol{\epsilon}}_t = \epsilon_\theta(\mathbf{z}_i^t, t, \mathbf{c}_i^0)$. The LDM is trained with a denoising score matching loss: $\mathcal{L}_{\text{diff}} = \mathbb{E}_{\mathbf{z}_0, t, \boldsymbol{\epsilon}} \|\boldsymbol{\epsilon} - \hat{\boldsymbol{\epsilon}}_t\|_2^2$, which ensures the predicted noise aligns with the true noise. By leveraging both temporal dynamics and our proposed neighborhood-aware conditioning, our approach achieves superior denoising and generates structurally coherent augmentations.

### 3.1.4 Task-Specific Augmentation Generation

D-GDA adapts its augmentation strategy to each task, using TSS, GVAE, and LDM to generate augmented samples.

**Node Classification:** For each TSS-selected target node $v_i$, we encode its one-hop neighborhood $\mathcal{N}(v_i)$ via the GVAE encoder to obtain the proposed conditioning vector $\mathbf{c}_i^0$ using mean aggregation. The LDM generates a new latent vector $\hat{\mathbf{z}}_{(\text{Aug},i)}^0$ conditioned on $\mathbf{c}_i^0$ which is then decoded into feature vector $\hat{\mathbf{x}}_{(\text{Aug},i)}$ using feature decoder of GVAE. The augmented node is linked with the most probable neighbor of $v_i$ as determined by the GVAE link predictor. The augmented node $v_{(\text{Aug},i)}$ is given the same label as $v_i$, and is added to the training set. Such neighborhood-aware augmentation allows better learning of the structurally hard nodes in the graph, ultimately enhancing generalization and robustness.

**Link Prediction:** TSS selects a set of target links each defined by its adjacent nodes $(v_i, v_j)$. For both of these nodes, we encode their 1-hop neighborhood nodes using GVAE encoder. An aggregated conditioning vector $\mathbf{c}_{ij}^0 = (\mathbf{c}_i^0 + \mathbf{c}_j^0)/2$ is computed by averaging across both neighborhoods. Using $\mathbf{c}_{ij}^0$ an augmented node latent vector $\mathbf{z}_{(Aug,i,j)}$ is estimated using LDM and decoded to feature $\mathbf{x}_{(Aug,i,j)}$ and is connected to both nodes $v_i$ and $v_j$. Such an augmentation guides the link prediction model to focus on challenging-to-learn links in a given graph.

**Graph Classification:** Given a set of graphs $\mathcal{G}$, TSS is employed to select a subset of target graphs for augmentation. For each selected graph $G_i$, the top-$\gamma$ highest degree nodes are identified as target nodes and for each of these nodes the augmentation procedure as discussed in the node classification section is followed. Selecting high-degree nodes ensures that the augmentation has a significant impact on the overall graph representation, as these nodes are more structurally influential. In contrast, augmenting low-degree nodes has minimal effect on the global structure, leading to less meaningful variation. Our augmented graph $G_{(\text{Aug},i)}$ inherits the label of its corresponding target graph $G_i$ and is added to the training set, thereby enriching data diversity and enhancing model generalization.

**Test-Time Augmentation:** The test-time augmentation has not been well explored in GDA [25]. Our proposed D-GDA being label-free is able to perform augmentation in test sets in node and graph classification tasks. Such augmentation facilitates efficient usage of graph structure resulting in performance improvement, without label assignment. During inference, we set each test-sample as a target and obtain its augmentation. The labels are predicted for both original and augmented samples. If different class labels are assigned, then the most confident label is selected as the final label.

Table 2: Node Classification performance comparison of D-GDA framework with SOTA methods on small-scale datasets. Best and 2nd best performances are in **bold** and underline, respectively.

| Method | Cora | Flickr | Citeseer | Pubmed | Mean |
|---|---|---|---|---|---|
| GCN [29] | 81.60±0.70 | 61.20±0.40 | 71.60±0.40 | 78.80±0.60 | 73.30 |
| +DropEdge [54] | 82.00±0.80 | 61.40±0.70 | 71.80±0.20 | 77.30±0.32 | 73.13 |
| +AdaEdge [7] | 81.90±0.70 | 61.20±0.50 | 72.80±0.70 | 77.40±0.50 | 73.33 |
| +NodeAug [80] | 82.10±0.90 | - | 71.40±0.60 | 78.80 ±0.40 | 77.43 |
| +GAug[90] | 83.60±0.50 | 62.20±0.30 | 73.30±1.10 | 80.20±0.30 | 74.83 |
| +Graph Mixup [73] | 73.80±0.02 | - | 64.30±0.04 | 76.60±0.18 | 71.57 |
| +FLAG [31] | 75.20±0.40 | 62.90±0.20 | 62.70±0.60 | 78.50±0.01 | 69.83 |
| +LA [42] | 84.60±0.50 | 64.24±0.30 | 74.70±0.50 | 81.70±0.70 | 76.31 |
| +NASA [3] | 85.10±0.30 | - | 75.50±0.40 | 80.20±0.30 | 80.30 |
| +DropMessage [13] | 83.33±0.11 | 53.55±0.23 | 71.83±0.09 | 79.20±0.06 | 71.98 |
| +S-Mixup [26] | 84.78±0.15 | - | 74.39±0.10 | 79.70±0.17 | 79.62 |
| +DropEdge++ [17] | 83.10±0.12 | 63.18±0.14 | 72.70±0.06 | 80.00±0.42 | 74.75 |
| +GraphPatcher [25] | 84.17±0.54 | - | 71.65±0.05 | 81.13±0.68 | 78.98 |
| +iGraphMix [23] | 83.78±0.42 | 53.61±0.12 | 73.67±0.61 | 79.93±0.60 | 72.75 |
| +SkipNode [43] | 82.00±0.40 | 50.73±0.09 | 69.60±0.50 | 77.50±0.70 | 69.96 |
| +GeoMix [91] | 84.08±0.74 | - | 75.06±0.36 | 80.06±0.93 | 79.73 |
| +RGDA [39] | 84.33±0.41 | - | 73.02±0.36 | 82.08±0.50 | 79.81 |
| +LeDA [41] | 78.60±0.32 | 62.64±0.32 | 67.50±0.18 | 79.70±0.02 | 72.11 |
| +DoG [71] | 84.00±0.30 | - | 73.60±0.40 | 82.80±0.30 | 80.13 |
| +D-GDA (ours) | **89.10±0.42** | **87.30±0.60** | **81.50±0.15** | **88.20±0.18** | **86.53** |

## 4 Experiments

**Experiments for Node Classification Task:** We evaluate D-GDA on four small-scale datasets: Cora, Citeseer, Pubmed [56], and Flickr [46] and two large-scale datasets: Ogbn-Arxiv [22], and Ogbn-Products [22] under transductive learning. Following [29, 7] we use semi-supervised settings for the small-scale datasets and following [22] supervised settings are used for the large-scale ones. D-GDA outperforms both the baseline **GCN** and compared SOTA methods across all datasets as shown in Tables 2 and 3. Compared to GCN, D-GDA improves test accuracy by 7.5% for Cora, 26.1% for Flickr, 9.9% for Citeseer, 9.4% for Pubmed, 3.18% for Ogbn-Arxiv, and 7.48% for Ogbn-Products. We also evaluate D-GDA on GNN backbones including **GAT** [67] and **GraphSAGE** [16] as shown in Table 4, where D-GDA again surpasses existing methods. On Cora, Citeseer, and Pubmed, improvements with GAT are 4.7%, 4.3%, and 9.1%, and with GraphSAGE are 5.3%, 7.5%, and 7.4%, respectively. More comparisons are shown in Appendix B.2. More details of all datasets are given in Appendix B.1.

**Experiments for Link Prediction Task:** We evaluate D-GDA on five link prediction benchmarks: ogbl-collab, ogbl-ddi, Cora, Citeseer, and Pubmed using GCN [29] and GSAGE [16] backbones and the results are shown in Table 5. D-GDA consistently outperforms existing methods on most datasets and backbones, with notable gains on smaller citation datasets. More results are shown in Appendix B.3.

Table 3: Node classification performance comparison on large-scale datasets.

| Method | Ogbn-Arxiv | Ogbn-Products |
|---|---|---|
| GCN [29] | 71.62±0.29 | 71.37±0.50 |
| +DropEdge [54] | 56.26±0.02 | 56.41±3.40 |
| +FLAG [31] | 71.75±0.01 | 76.14±0.30 |
| +LA [42] | 72.08±0.14 | 76.11±0.09 |
| +DropMessage [13] | 71.27±0.02 | - |
| +RGDA [39] | 72.88±0.65 | - |
| +DoG [71] | 73.10±0.30 | - |
| +D-GDA (ours) | **74.80±0.04** | **78.85±0.24** |

Table 4: Node Classification Performance of D-GDA on GAT [67] and GSAGE [16] backbones.

| | Method | Cora | Citeseer | Pubmed |
|---|---|---|---|---|
| GAT | Vanilla | 81.3 | 70.5 | 79.4 |
| | +GAug [90] | 82.2 | 71.6 | 79.3 |
| | +LA [42] | 84.7 | 74.7 | 79.8 |
| | +SkipNode [43] | 81.6 | 68.4 | 77.6 |
| | +D-GDA (ours) | **89.4** | **79.0** | **88.9** |
| GSAGE | Vanilla | 81.3 | 70.6 | 76.8 |
| | +GAug [90] | 83.2 | 72.7 | 78.5 |
| | +SkipNode [43] | 81.5 | 68.5 | 77.4 |
| | +RGDA [39] | 83.4 | 72.6 | 82.1 |
| | +D-GDA (ours) | **88.7** | **80.2** | **89.5** |

Table 5: Link prediction performance comparison with SOTA methods.

| | Method | Ogbl-collab (Hits@50) | Ogbl-ddi (Hits@20) | Cora (AUC) | Citeseer (AUC) | Pubmed (AUC) |
|---|---|---|---|---|---|---|
| GCN | Vanilla | 44.75±1.07 | 37.07±5.07 | 89.55±0.53 | 69.47±1.40 | 96.11±0.80 |
| | +FLAG [31] | 46.22±0.81 | 51.41±3.76 | 91.34±0.25 | 85.63±0.73 | 96.21±0.52 |
| | +CFLP [89] | - | 52.51±1.09 | 92.55±0.50 | 89.65±0.20 | 96.99±0.08 |
| | +GM [63] | - | **59.18±2.09** | 92.02±0.45 | 90.61±0.30 | 97.35±0.15 |
| | +D-GDA (ours) | **50.24±0.97** | 57.53±0.15 | **96.96±0.15** | **93.72±0.36** | **97.82±0.18** |
| GSAGE | Vanilla | 48.10±0.81 | 53.90±4.74 | 91.14±0.34 | 86.98±1.39 | 96.78±0.11 |
| | +FLAG [31] | 48.44±0.40 | 63.31±6.06 | 91.52±0.42 | 90.48±0.19 | 96.89±0.14 |
| | +CFLP [89] | - | 75.49±4.33 | 92.61±0.52 | 91.84±0.20 | 97.01±0.01 |
| | +GM [63] | - | **79.26±1.12** | 92.41±0.50 | 92.85±0.35 | 98.62±0.09 |
| | +D-GDA (ours) | **51.08±0.72** | 78.82±0.35 | **97.62±0.35** | **95.64±0.12** | **98.88±0.32** |

Table 6: Graph Classification Performance (ROC-AUC) comparison with SOTA methods.

| | Method | ogbg-molSIDER | ogbg-molClinTox | ogbg-molHIV | ogbg-molBACE |
|---|---|---|---|---|---|
| GCN | Vanilla [29] | 59.60±1.17 | 88.55±2.09 | 76.06±0.97 | 71.47±0.32 |
| | +FLAG [31] | 59.92±0.05 | 92.06±0.32 | 76.83±1.01 | 74.28±1.16 |
| | +RGDA [39] | 63.16±1.51 | 89.61±1.50 | 77.94±0.65 | 81.91±2.41 |
| | +D-GDA (ours) | **64.85±0.32** | **94.82±0.12** | **79.03±0.92** | **83.45±0.85** |
| GIN | Vanilla | 58.10±0.90 | 88.80±3.80 | 75.58±1.40 | 77.50±2.80 |
| | +FLAG [31] | 60.74±0.07 | 87.75±0.16 | 76.54±1.14 | 79.10±1.20 |
| | +G-Mixup [18] | 56.80±3.50 | 60.20±7.50 | 77.10±1.10 | 77.80±3.30 |
| | +DCT [40] | 63.90±0.30 | 92.10±0.80 | **79.50±1.00** | 85.60±0.60 |
| | +RGDA [39] | 60.14±2.04 | 87.89±3.68 | 79.32±0.92 | 82.37±2.37 |
| | +D-GDA (ours) | **64.83±0.14** | **94.72±0.48** | 79.45±1.37 | **86.12±0.98** |

**Experiments for Graph Classification Task:** We evaluate D-GDA on four graph classification benchmarks: ogbg-molSIDER, ogbg-molClinTox, ogbg-molHIV, and ogbg-molBACE, using GCN [29] and GIN [79] backbones. D-GDA outperforms compared SOTA methods on most datasets and backbones as shown in Table 6. Additional details are given in Appendix B.4.

**Experiments on Long Tailed Datasets:** We evaluate D-GDA on two long-tailed datasets, Cora-LT and Citeseer-LT, with an imbalance ratio of $\rho = 100$ to assess its performance under class imbalance. Unlike prior methods that focus on oversampling minority classes, D-GDA adopts a class-agnostic strategy by identifying and selectively augmenting challenging samples. As shown in Table 7, D-GDA achieves consistent improvements on both datasets without requiring imbalance-specific tuning, highlighting its effectiveness in long-tailed settings. Additional results are provided in Appendix B.5.

**Improvements in ML Safety Measures.** We evaluate D-GDA on three ML safety measures: calibration (assessing how well predicted probabilities reflect actual correctness), corruption (robustness to input perturbations), and consistency (stability of predictions under minor input changes). Models are trained on clean data and tested across these tasks. D-GDA shows strong performance on all metrics (see Table 8). More details are in Appendix B.6.

Table 7: Balanced accuracy (%) comparison under class imbalance ($\rho = 100$).

| Method | Cora-LT | Citeseer-LT |
|---|---|---|
| GCN (Vanilla) | 59.42±0.74 | 44.64±0.42 |
| +ReNode [8] | 67.61±0.13 | 47.78±0.31 |
| +GraphSMOTE [87] | 66.29±0.43 | 44.40±0.29 |
| +GraphENS [49] | 70.31±0.24 | 55.42±0.35 |
| +TAM (G-ENS) [59] | 72.10±0.23 | 57.15±0.34 |
| +GraphSHA [37] | 74.62±0.29 | 59.04±0.41 |
| +GraphVCM [52] | 75.81±0.42 | 60.53±1.37 |
| +D-GDA (ours) | **80.34±0.51** | **64.95±0.18** |

**Adversarial Robustness.** We evaluate D-GDA robustness against four evasion attacks: Random (randomly flip edges), DICE [74] (deletes intra-class edges and adds inter-class ones), GFAttack [6] (optimizes a low-rank loss for structural perturbations), and Meta-attack [93] (meta-gradient-based loss maximization), at two perturbation ratios ($\sigma \in 0.05, 0.2$). Table 9 shows improved node classification accuracy under these attacks compared to baseline GCN. More details are in Appendix B.7.

**D-GDA promotes flatter minima for better generalization.** Following [86, 85], we analyze whether D-GDA leads to flatter loss landscapes. Using a trained GCN, we compare D-GDA to DropEdge, FLAG, and LA under Gaussian noise perturbations of model parameters. As shown in Figure 3, D-GDA maintains higher accuracy and lower loss, indicating flatter minima. t-SNE

Table 8: D-GDA performance comparison on ML safety measures (lower is better). Error is reported on clean data, corruptions are Gaussian (G), Shot (S), Impulse (I), and Shift noise.

| | Method | Error | Calib. | Consist. | Corruptions | | | | Mean |
|---|---|---|---|---|---|---|---|---|---|
| | | | | | G | S | I | Shift | |
| Cora | GCN [29] | 18.40 | 19.65 | 16.26 | 49.42 | 57.12 | 54.00 | 32.60 | 35.06 |
| | +DropEdge [54] | 18.00 | 21.16 | 18.74 | 62.62 | 20.90 | 71.08 | 19.00 | 35.38 |
| | +FLAG [31] | 24.80 | 19.52 | 25.59 | 33.71 | 35.17 | 37.48 | 31.7 | 28.41 |
| | +LA [42] | 15.40 | 34.33 | 27.35 | 63.42 | 27.68 | 78.2 | 27.1 | 34.47 |
| | +D-GDA (ours) | **10.9** | **9.96** | **15.77** | **13.34** | **12.81** | **16.64** | **13.59** | **13.29** |
| Citeseer | GCN [29] | 28.40 | 33.43 | 25.42 | 46.84 | 49.90 | 51.16 | 40.20 | 39.34 |
| | +DropEdge [54] | 28.20 | 24.14 | 19.83 | 70.36 | 30.9 | 75.66 | 29.99 | 40.38 |
| | +FLAG [31] | 37.30 | 27.95 | 25.48 | 39.08 | 61.6 | 56.58 | 39.7 | 38.45 |
| | +LA [42] | 25.30 | 31.44 | 26.36 | 68.38 | 27.98 | 78.86 | 26.8 | 36.44 |
| | +D-GDA (ours) | **18.5** | **11.51** | **18.6** | **21.29** | **20.52** | **24.28** | **20.79** | **19.36** |
| Pubmed | GCN [29] | 21.2 | 16.74 | 11.03 | 45.88 | 31.26 | 51.86 | 33.30 | 30.18 |
| | +DropEdge [54] | 22.7 | 17.47 | 14.28 | 50.32 | 23.64 | 55.82 | 21.09 | 29.33 |
| | +FLAG [31] | 21.5 | 21.05 | 9.11 | 39.64 | 35.18 | 59.92 | 24.29 | 30.09 |
| | +LA [42] | 18.3 | 16.92 | 10.24 | 36.00 | 19.34 | 46.32 | 19.30 | 23.77 |
| | +D-GDA (ours) | **11.8** | **8.17** | **8.13** | **23.8** | **15.44** | **35.72** | **14.7** | **16.82** |

Table 9: Results on adversarial robustness.

| Datasets | Method | Clean | Random | | DICE | | GF-Attack | | Meta-Attack | |
|---|---|---|---|---|---|---|---|---|---|---|
| | | 0.00 | 0.05 | 0.2 | 0.05 | 0.2 | 0.05 | 0.2 | 0.05 | 0.2 |
| Cora | GCN | 81.6 | 78.7 | 77.2 | 78.2 | 74.5 | 79.8 | 78.9 | 79.1 | 78.4 |
| | +D-GDA | 89.1 | 85.8 | 83.3 | 85.7 | 82.3 | 85.3 | 83.9 | 85.1 | 83.9 |
| Citeseer | GCN | 71.6 | 64.1 | 62.6 | 63.42 | 60.9 | 64.9 | 54.51 | 63.6 | 55.13 |
| | +D-GDA | 81.5 | 77.9 | 73.7 | 77.3 | 73.4 | 77.8 | 72.3 | 77.5 | 73.1 |
| Pubmed | GCN | 78.8 | 77.9 | 76 | 77.6 | 72.7 | 77.9 | 68.6 | 76.1 | 68.9 |
| | +D-GDA | 88.2 | 84.4 | 83.1 | 83.9 | 81.9 | 84.2 | 82.5 | 84.8 | 82.9 |

visualizations (Figure 4) show more cohesive, well-separated clusters with D-GDA, further supported by an improved silhouette score (from 0.02 to 0.38). See Appendix B.9 for details.

**Over-Smoothness Analysis.** Using MADGap [7] by measuring feature similarity gaps between neighboring and remote nodes: Table 10 shows that D-GDA consistently achieves higher MADGap scores than others (see Appendix B.9 for details).

**Ablation Study.** Table 11 shows that model performance improves as augmentation is applied progressively to the training, validation, and test sets, with the best accuracy achieved when all are aug-

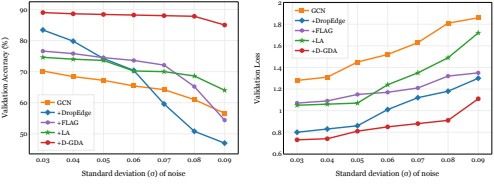

(a) Validation Accuracy     (b) Validation Loss

Figure 3: D-GDA encourages flatter minima.

mented. Validation-set augmentation adds unlabeled synthetic nodes to aid learning in difficult regions, while test-set augmentation serves as test-time augmentation by combining predictions from original and augmented nodes. To evaluate the **role of the GVAE Link Predictor** in guiding augmented node connections, we compare it with three heuristics: Random (target or 1-hop neighbor), Target-only, and Target+1-hop. Table 12 shows that our method outperforms all heuristics, highlighting its effectiveness in capturing meaningful structure and boosting classification accuracy. To assess the **impact of GVAE Feature Decoder**, we replaced it with two heuristics: random features (Rand Feat) and target node features (Target Feat). Both led to notable performance drops (Table 13),

Table 10: MADGap Comparison.

| Method | Cora | Citeseer | Pubmed |
|---|---|---|---|
| GCN (Vanilla) | 0.49 | 0.53 | 0.71 |
| +DropEdge | 0.41 | 0.49 | 0.27 |
| +AdaEdge | **0.66** | 0.53 | 0.73 |
| +FLAG | 0.12 | 0.17 | 0.31 |
| +LA | 0.49 | 0.44 | 0.35 |
| +D-GDA | 0.64 | **0.81** | **0.86** |

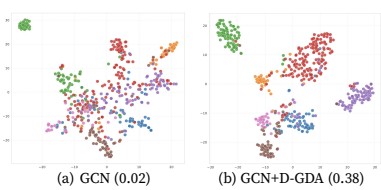

(a) GCN (0.02)     (b) GCN+D-GDA (0.38)

Figure 4: t-SNE of node embeddings on Cora.

Table 11: Comparison of augmentation strategies.

| Augmentation | | | Cora | Citeseer | Pubmed |
|---|---|---|---|---|---|
| Tr | Val | Test | | | |
| ✓ | | | 84.9 | 75.7 | 80.7 |
| ✓ | | ✓ | 85.8 | 77.3 | 83.5 |
| ✓ | ✓ | | 88.2 | 79.8 | 85.5 |
| ✓ | ✓ | ✓ | 89.1 | 81.5 | 88.2 |

Table 12: Importance of proposed link predictor.

| Edge injection | Cora | Citeseer | Pubmed |
|---|---|---|---|
| Random | 82.1 | 72.5 | 79.4 |
| Target | 85.3 | 75.3 | 80.8 |
| 1-hop | 83.7 | 74.5 | 79.6 |
| Ours | 89.1 | 81.5 | 88.2 |

Table 13: Importance of Feature Decoder

| Node Features | Cora | Citeseer | Pubmed |
|---|---|---|---|
| Rand Feat | 78.6 | 69.92 | 75.94 |
| Target Feat | 78.8 | 78.9 | 71.95 |
| Ours | 89.1 | 81.5 | 88.2 |

Table 14: Importance of each module in D-GDA

| TSS | LDM | GVAE | GAE | Cora | Citeseer | Pubmed |
|---|---|---|---|---|---|---|
| ✓ | ✓ | ✓ | × | 89.1 | 81.5 | 88.2 |
| × | ✓ | ✓ | × | 83.5 | 72.7 | 80.9 |
| ✓ | × | ✓ | × | 83.8 | 73.5 | 81.5 |
| ✓ | ✓ | × | ✓ | 87.3 | 79.6 | 85.8 |

Table 15: Training time and inference time per sample (sec) comparison of D-GDA with baselines.

| Type | Method | Cora | | Flickr | | Ogbn-Arxiv | |
|---|---|---|---|---|---|---|---|
| | | Train | Inf. | Train | Inf. | Train | Inf. |
| **Non-Diffusion** | **GCN** [29] | 9.05 | 0.003 | 79.82 | 0.058 | 928.21 | 0.567 |
| | **DropEdge**[54] | 11.75 | 0.005 | 132.23 | 0.096 | 1786.40 | 0.813 |
| | **FLAG** [31] | 34.20 | 0.019 | 547.80 | 0.160 | 3307.50 | 0.815 |
| **Diffusion** | **D-GDA** w/o TTA (ours) | 37.72 | 0.019 | 328.17 | 0.160 | 2507.20 | 0.816 |
| | **D-GDA** with TTA (ours) | 37.72 | 1.41 | 328.17 | 1.33 | 2507.20 | 1.80 |

highlighting the decoder's role in generating meaningful features and enhancing D-GDA effectiveness. Table 14 assesses the **contributions of each module in the D-GDA** framework. The full model (TSS + GVAE + LDM) achieves the best performance. Removing TSS leads to a performance drop, highlighting its role in directing augmentation to challenging regions. Replacing LDM with GVAE results in subpar performance, underscoring LDM's critical role in generating synthetic nodes and edges. Substituting GVAE with GAE also produces lower results. For more ablations see Appendix B.10.

**Computational Cost Comparison.** Table 15, compares the overall training and inference times (in seconds) of D-GDA with and w/o TTA against baseline GCN, and non-diffusion graph data augmentation methods including DropEdge, and FLAG, across three datasets of varying sizes: Cora, Flickr, and Ogbn-Arxiv (large-scale dataset). Note that the execution times of SOTA Diffusion based GDA methods, could not be compared due to non-availability of their codes. Among these methods, DropEdge consistently demonstrates the lowest training and inference costs, making it the most efficient overall. D-GDA shows better efficiency than FLAG on the larger datasets, Flickr and Ogbn-Arxiv, highlighting its scalability. Additionally, the D-GDA variant without TTA achieves inference times comparable to FLAG. See Appendix B.11 for component-wise training time of D-GDA.

## 5 Conclusions

In this work, we proposed Diffusion-based Graph Data Augmentation (D-GDA), which leverages LDMs to enhance GCN performance. Target Sample Selector (TSS) is introduced for identifying challenging-to-learn samples. Graph Variational Autoencoder (GVAE) is employed for learning compact latent representations. We propose neighbourhood based constraint for Latent Diffusion Model (LDM) based node generation. D-GDA enables label-free augmentation applicable to node, edge, and graph-level tasks. Extensive experiments on 14 benchmarks demonstrate D-GDA superiority over the compared SOTA techniques. D-GDA also improves safety measures, including calibration, corruption, consistency, and robustness to adversarial attacks. These results highlight D-GDA's potential as an effective and versatile augmentation strategy for advancing graph learning.

## Acknowledgement

This work was supported by the International Science and Technology Cooperation Project of Guangzhou Economic and Technological Development District (No.2023GH16), China.

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

# Diffusion-Guided Graph Data Augmentation

## Supplementary Material

## A    How Proposed D-GDA is Different from the Existing SOTA Methods?

Table 16 presents an extended comparison of D-GDA against 30 State-Of-The-Art (SOTA) methods, evaluating key aspects such as basic building blocks, supported tasks, robustness assessments, learning settings, and test-time augmentation. It table is an extension of Table 1 in the main paper.

Table 16: (Table 1 extension) A comparison of SOTA GDA methods: 1) **Basic building blocks:** graph auto-encoder (GAE), Graph Variational auto-encoder (GVAE), and Diffusion models. 2) **Supported tasks:** Node Classification (NC), Link Prediction (LP), and Graph Classification (GC). 3) **Robustness evaluations:** adversarial robustness (Adv) and machine learning safety measures (MSM). 4) **Learning Settings:** semi-supervised, Supervised, and Long-tailed (LT). 5) **Test-Time Augmentation** (TTA).

| Methods | Basic Building Blocks | | | Supported Tasks | | | Robustness | | Learning Setting | | | TTA |
|---|---|---|---|---|---|---|---|---|---|---|---|---|
| | GAE | GVAE | Diff. | NC | LP | GC | Adv | MSM | Semi | Sup. | LT | |
| DropEdge [54] | ✗ | ✗ | ✗ | ✓ | ✗ | ✗ | ✗ | ✗ | ✓ | ✗ | ✗ | ✗ |
| AdaEdge [7] | ✗ | ✗ | ✗ | ✓ | ✗ | ✗ | ✗ | ✗ | ✓ | ✗ | ✗ | ✗ |
| NodeAug [80] | ✗ | ✗ | ✗ | ✓ | ✗ | ✗ | ✗ | ✗ | ✓ | ✗ | ✗ | ✗ |
| GAug [90] | ✓ | ✗ | ✗ | ✓ | ✗ | ✗ | ✗ | ✗ | ✓ | ✗ | ✗ | ✗ |
| Graph Mixup [73] | ✗ | ✗ | ✗ | ✓ | ✗ | ✓ | ✗ | ✗ | ✓ | ✓ | ✗ | ✗ |
| ReNode [8] | ✗ | ✗ | ✗ | ✓ | ✗ | ✗ | ✗ | ✗ | ✗ | ✗ | ✓ | ✗ |
| GraphSMOTE [87] | ✗ | ✗ | ✗ | ✓ | ✗ | ✗ | ✗ | ✗ | ✗ | ✗ | ✓ | ✗ |
| SR+DR [60] | ✗ | ✗ | ✗ | ✓ | ✗ | ✗ | ✗ | ✗ | ✓ | ✗ | ✗ | ✗ |
| GraphENS [49] | ✗ | ✗ | ✗ | ✓ | ✗ | ✗ | ✗ | ✗ | ✓ | ✗ | ✓ | ✗ |
| FLAG [31] | ✗ | ✗ | ✗ | ✓ | ✓ | ✓ | ✗ | ✗ | ✓ | ✓ | ✗ | ✗ |
| LA [42] | ✗ | ✓ | ✗ | ✓ | ✓ | ✓ | ✗ | ✗ | ✓ | ✓ | ✗ | ✗ |
| NASA [3] | ✗ | ✗ | ✗ | ✓ | ✗ | ✗ | ✗ | ✗ | ✓ | ✗ | ✗ | ✗ |
| CFLP [89] | ✗ | ✗ | ✗ | ✗ | ✓ | ✗ | ✗ | ✗ | ✗ | ✓ | ✗ | ✗ |
| TAM(G-ENS) [59] | ✗ | ✗ | ✗ | ✓ | ✗ | ✗ | ✗ | ✗ | ✗ | ✗ | ✓ | ✗ |
| G-Mixup [18] | ✗ | ✗ | ✗ | ✗ | ✗ | ✓ | ✗ | ✗ | ✗ | ✗ | ✓ | ✗ |
| GraphSHA [37] | ✗ | ✗ | ✗ | ✓ | ✗ | ✗ | ✗ | ✗ | ✗ | ✗ | ✓ | ✗ |
| DropMessage [13] | ✗ | ✗ | ✗ | ✓ | ✓ | ✗ | ✗ | ✗ | ✓ | ✓ | ✗ | ✗ |
| S-Mixup [26] | ✗ | ✗ | ✗ | ✓ | ✗ | ✗ | ✗ | ✗ | ✓ | ✗ | ✗ | ✗ |
| DropEdge++ [17] | ✗ | ✗ | ✗ | ✓ | ✗ | ✗ | ✗ | ✗ | ✓ | ✓ | ✗ | ✗ |
| GraphPatcher [25] | ✗ | ✗ | ✗ | ✓ | ✗ | ✗ | ✗ | ✗ | ✓ | ✗ | ✗ | ✓ |
| iGraphMix [23] | ✗ | ✗ | ✗ | ✓ | ✓ | ✗ | ✗ | ✗ | ✓ | ✓ | ✗ | ✗ |
| SkipNode [43] | ✗ | ✗ | ✗ | ✓ | ✗ | ✗ | ✗ | ✗ | ✓ | ✓ | ✗ | ✗ |
| GeoMix [91] | ✗ | ✗ | ✗ | ✓ | ✗ | ✗ | ✗ | ✗ | ✓ | ✗ | ✗ | ✗ |
| RGDA [39] | ✗ | ✗ | ✗ | ✓ | ✗ | ✓ | ✗ | ✗ | ✓ | ✓ | ✗ | ✗ |
| xAI-DropEdge [11] | ✗ | ✗ | ✗ | ✓ | ✗ | ✗ | ✗ | ✗ | ✓ | ✗ | ✗ | ✗ |
| DCT [40] | ✗ | ✗ | ✓ | ✗ | ✗ | ✓ | ✗ | ✗ | ✗ | ✓ | ✗ | ✗ |
| LeDA [41] | ✗ | ✗ | ✗ | ✓ | ✗ | ✗ | ✓ | ✗ | ✓ | ✗ | ✗ | ✗ |
| GM [63] | ✗ | ✗ | ✗ | ✗ | ✓ | ✗ | ✗ | ✗ | ✗ | ✓ | ✗ | ✗ |
| DoG [71] | ✓ | ✗ | ✓ | ✓ | ✗ | ✗ | ✗ | ✗ | ✓ | ✗ | ✗ | ✗ |
| GraphVCM [52] | ✗ | ✗ | ✗ | ✓ | ✗ | ✗ | ✗ | ✗ | ✗ | ✗ | ✓ | ✗ |
| D-GDA (ours) | ✗ | ✓ | ✓ | ✓ | ✓ | ✓ | ✓ | ✓ | ✓ | ✓ | ✓ | ✓ |

## B    Details of Experiments Performed using Proposed D-GDA

### B.1    Datasets' Details

**Node Classification:** We evaluate our proposed method on four small-scale and two large-scale datasets. Cora, Citeseer, and Pubmed [56] are citation networks where nodes are papers and edges represent citation links. Node features are bag-of-words representation of papers. Flickr [46] is a social network in which nodes are users and edges represent user interaction from an image and video hosting website. OGBN-Arxiv [22] is a directed graph, representing the citation network between all Computer Science (CS) arXiv papers. Each node is an arXiv paper and each directed edge indicates that one paper cites another one. Each paper comes with a 128-dimensional feature vector obtained by

averaging the embeddings of words in its title and abstract. OGBN-Products [22] is a co-purchasing network having products as nodes and the edge indicates that the products are bought together. Table 17 summarizes the dataset statistics.

Table 17: Summary statistics of node classification evaluation datasets

| Datsets | #Nodes | #Edges | #Classes | Train/Valid/Test split |
|---------|--------|--------|----------|------------------------|
| **Cora** | 2,708 | 5,278 | 7 | 140/500/1,000 [29] |
| **Citeseer** | 3,327 | 4,552 | 6 | 120/500/1,000 [29] |
| **Flickr** | 7,575 | 2,39,738 | 9 | 757/1,515/5,303 [90] |
| **Pubmed** | 19,717 | 44,338 | 3 | 60/500/1,000 [29] |
| **OGBN-Arxiv** | 169,343 | 1,166,243 | 40 | 90,941/29,799/48,603 [22] |
| **OGBN-Products** | 2,449,029 | 61,859,140 | 47 | 196,615/39,323/2,213,091 [22] |

**Link Prediction:** We evaluate our proposed method on five datasets: ogbl-collab [70] ogbl-ddi [75], Cora, Citeseer, and Pubmed [56]. Ogbl-collab [70] is an undirected collaboration network between authors indexed by MAG. Each node represents an author, and edges indicate collaborations between authors. All nodes have 128-dimensional features, obtained by averaging the word embeddings of papers published by the authors. Each edge is associated with two pieces of meta-information: the year and the edge weight, representing the number of co-authored papers published in that year. The task is to predict future author collaborations given past collaborations. Ogbl-ddi [75] is an undirected graph representing the drug-drug interaction network. Each node represents an FDA-approved or experimental drug, and edges represent interactions between drugs, indicating that the joint effect of taking the two drugs together is significantly different from their independent effects. The task is to predict drug-drug interactions given information on known interactions. Cora, Citeseer, and Pubmed are citation networks. Table 18 summarizes the dataset details.

Table 18: Summary Statistics of link prediction evaluation datasets

| Datasets | #Nodes | #Edges | #Features | Train/Val/Test |
|----------|--------|--------|-----------|----------------|
| Ogbl-collab | 235,868 | 1,285,465 | 128 | 92/04/04 [22] |
| Ogbl-ddi | 4,267 | 1,334,889 | - | 80/10/10 [22] |
| Cora | 2,708 | 5,278 | 1,433 | 85/05/10 [35] |
| Citeseer | 3,327 | 4,552 | 3,703 | 85/05/10 [35] |
| Pubmed | 19,717 | 44,338 | 500 | 85/05/10 [35] |

**Graph Classification:** We evaluate our proposed method on four molecular property prediction datasets: ogbg-molSIDER, ogbg-ClinTox, ogbg-molHIV, and ogbg-molBACE [78]. Each graph represents a molecule, where nodes are atoms, and edges are chemical bonds. The input node features are 9-dimensional, including atomic number, chirality, and additional atom features such as formal charge and ring membership. Table 19 summarizes the key statistics for all five datasets.

Table 19: Summary Statistics of graph classification evaluation datasets

| Datasets | #Graphs | Average #Nodes | Average #Edges | #Classes | Train/Val/Test |
|----------|---------|----------------|----------------|----------|----------------|
| Ogbg-molSIDER | 1,427 | 33.6 | 70.7 | 27 | 80/10/10 [22] |
| Ogbg-molClinTox | 1,477 | 26.2 | 55.8 | 2 | 80/10/10 [22] |
| Ogbg-molHIV | 41,127 | 25.5 | 27.5 | 2 | 80/10/10 [22] |
| Ogbg-molBACE | 1,513 | 34.1 | 73.7 | 2 | 80/10/10 [22] |

**Datasets for Class Imbalance Evaluations:** We evaluate our method on the Cora and Citeseer datasets [56] in a long-tailed setting. To assess models under conditions of high class imbalance, we create long-tailed citation networks following the methodology outlined in [10]. This involves adjusting the class distribution to follow a long-tailed pattern by systematically removing nodes, thereby increasing the imbalance ratio, which is defined as the ratio between the most frequent class and the least frequent class. To achieve this, we sort the classes in descending order by size and iteratively remove nodes from each class, starting with the major class. During the node removal process, we prioritize eliminating nodes with low degrees and removing the corresponding edges while striving to maintain graph connectivity. Table 20 summarizes the dataset statistics.

Table 20: Summary Statistics of Graph Class Imbalance Evaluation Datasets

| Datasets | #Nodes | #Edges | #Classes | #Training Nodes Per Class |
|---|---|---|---|---|
| **Cora** | 2,708 | 5,278 | 7 | [34, 7, 158, 341, 73, 15, 3] |
| **Citeseer** | 3,327 | 4,552 | 6 | [3, 23, 371, 147, 58, 9] |

Table 21: (Table 2 extension) Node classification performance comparison of D-GDA variants and SOTA methods on small-scale datasets. Best and $2^{nd}$ best performances are in **bold** and underline, respectively.

| Method | Cora | Flickr | Citeseer | Pubmed | Mean |
|---|---|---|---|---|---|
| GCN [29] | 81.60±0.70 | 61.20±0.40 | 71.60±0.40 | 78.80±0.60 | 73.30 |
| +DropEdge [54] | 82.00±0.80 | 61.40±0.70 | 71.80±0.20 | 77.30±0.32 | 73.13 |
| +AdaEdge [7] | 81.90±0.70 | 61.20±0.50 | 72.80±0.70 | 77.40±0.50 | 73.33 |
| +SR+DR [60] | 83.44±0.34 | - | 71.76±0.15 | 80.64±0.12 | 78.61 |
| +NodeAug [80] | 82.10±0.90 | - | 71.40±0.60 | 78.80 ±0.40 | 77.43 |
| +GAug[90] | 83.60±0.50 | 62.20±0.30 | 73.30±1.10 | 80.20±0.30 | 74.83 |
| +Graph Mixup [73] | 73.80±0.02 | - | 64.30±0.04 | 76.60±0.18 | 71.57 |
| +FLAG [31] | 75.20±0.40 | 62.90±0.20 | 62.70±0.60 | 78.50±0.01 | 69.83 |
| +LA [42] | 84.60±0.50 | 64.24±0.30 | 74.70±0.50 | 81.70±0.70 | 76.31 |
| +NASA [3] | 85.10±0.30 | - | 75.50±0.40 | 80.20±0.30 | 80.30 |
| +DropMessage [13] | 83.33±0.11 | 53.55±0.23 | 71.83±0.09 | 79.20±0.06 | 71.98 |
| +S-Mixup [26] | 84.78±0.15 | - | 74.39±0.10 | 79.70±0.17 | 79.62 |
| +DropEdge++ [17] | 83.10±0.12 | 63.18±0.14 | 72.70±0.06 | 80.00±0.42 | 74.75 |
| +GraphPatcher [25] | 84.17±0.54 | - | 71.65±0.05 | 81.13±0.68 | 78.98 |
| +xAI-DropEdge [11] | 83.60±0.50 | - | 74.00±0.40 | 79.50±0.40 | 79.03 |
| +iGraphMix [23] | 83.78±0.42 | 53.61±0.12 | 73.67±0.61 | 79.93±0.60 | 72.75 |
| +SkipNode [43] | 82.00±0.40 | 50.73±0.09 | 69.60±0.50 | 77.50±0.70 | 69.96 |
| +GeoMix [91] | 84.08±0.74 | - | 75.06±0.36 | 80.06±0.93 | 79.73 |
| +RGDA [39] | 84.33±0.41 | - | 73.02±0.36 | 82.08±0.50 | 79.81 |
| +LeDA [41] | 78.60±0.32 | 62.64±0.32 | 67.50±0.18 | 79.70±0.02 | 72.11 |
| +DoG [71] | 84.00±0.30 | - | 73.60±0.40 | 82.80±0.30 | 80.13 |
| +D-GDA (ours) | **89.10±0.42** | **87.30±0.60** | **81.50±0.15** | **88.20±0.18** | **86.53** |

## B.2 More Details of Experiments for Node Classification

### B.2.1 Experiment Setup and Implementation Details

For TSS, we train a baseline 2-layer Graph Convolutional Network (GCN) with a hidden dimension of 32, trained using the Adam optimizer with a learning rate of 0.001 for 500 epochs. Early stopping is applied with a patience of 20 epochs to prevent overfitting. Next, we train a Graph Variational Autoencoder (GVAE) to learn latent representation. The GVAE consists of a 2-layer GCN encoder to learn node representations and two 2-layer Multi-Layer Perceptrons (MLPs) for decoding edge and feature information, respectively. We set the latent dimension to 64. The GVAE is optimized using a composite loss function: $\mathcal{L}_{GVAE} = \mathcal{L}_{edge} + \lambda_1 \mathcal{L}_{feat} + \lambda_2 \mathcal{L}_{KL}$, where $\lambda_1 = 0.3$ weights the feature reconstruction loss and $\lambda_2 = 0.01$ controls the KL-divergence term for regularization as recommended by the original authors [30]. The GVAE is trained for 1000 epochs with the Adam optimizer, using a learning rate of 0.01 and a weight decay of $5 \times 10^{-5}$. Following [34, 47], we apply edge masking at a rate of 0.3 and feature masking at a rate of 0.5 to enhance robustness during training. Finally, we train a Latent Diffusion Model (LDM) with a timestep $T = 1000$ and a hidden dimension of 64. The LDM is optimized using the Adam optimizer with a learning rate of $1 \times 10^{-4}$ for 1000 epochs, generating high-quality augmented samples for the node classification task.

### B.2.2 Improvements over GCN backbone

Table 21 presents a comprehensive comparison of D-GDA against existing SOTA methods using the GCN backbone. Across all evaluated datasets, D-GDA consistently outperforms both the baseline GCN and the second-best SOTA method, demonstrating significant performance improvements.

### B.2.3 Improvements over GAT backbone

We evaluate the performance of D-GDA on the Graph Attention Network (GAT) backbone [67] using three benchmark citation network datasets: Cora, Citeseer, and Pubmed. Table 22 presents a comparative analysis of D-GDA against existing SOTA methods under the GAT architecture. D-GDA achieves substantial performance gains over both the baseline GAT model and current SOTA approaches. Specifically, D-GDA outperforms the GAT baseline by 8.1%, 8.5%, and 9.5%, and exceeds the previous SOTA performance by 4.7%, 4.3%, and 9.1% on Cora, Citeseer, and Pubmed, respectively. Given that GAT is a self-attention-based model, where performance is known to be highly sensitive to both graph connectivity and node features, the observed improvements highlight the effectiveness and compatibility of the augmentation strategies introduced by D-GDA within attention-based architectures.

Table 22: (Table 4 extension) Node classification performance comparison of D-GDA using additional backbones including GAT [67] and GSAGE [16].

|  | Method | Cora | Citeseer | Pubmed |
|---|---|---|---|---|
| GAT | Vanilla | 81.3 | 70.5 | 79.4 |
|  | +DropEdge [54] | 81.9 | 71.0 | 79.6 |
|  | +AdaEdge [7] | 82.0 | 71.1 | 76.6 |
|  | +SR+DR [60] | 83.5 | 72.4 | 79.4 |
|  | +NodeAug [80] | 84.1 | 70.8 | 78.3 |
|  | +GAug [90] | 82.2 | 71.6 | 79.3 |
|  | +LA [42] | _84.7_ | _74.7_ | _79.8_ |
|  | +DropMessage [13] | 82.2 | 71.5 | 78.1 |
|  | +xAI-DropEdge [11] | 82.6 | 72.8 | 78.8 |
|  | +iGraphMix [23] | 83.2 | 72.3 | 78.4 |
|  | +LeDA [41] | 83.5 | 72.1 | 79.5 |
|  | +SkipNode [43] | 81.6 | 68.4 | 77.6 |
|  | +D-GDA (ours) | **89.4** | **79.0** | **88.9** |
| GSAGE | Vanilla | 81.3 | 70.6 | 76.8 |
|  | +DropEdge [54] | 81.6 | 70.8 | 77.1 |
|  | +AdaEdge [7] | 81.5 | 71.3 | 77.2 |
|  | +NodeAug [80] | 82.2 | 70.2 | 78.1 |
|  | +GAug [90] | 83.2 | _72.7_ | 78.5 |
|  | +LeDA [41] | 82.2 | 72.3 | 77.6 |
|  | +SkipNode [43] | 81.5 | 68.5 | 77.4 |
|  | +RGDA [39] | _83.4_ | 72.6 | _82.1_ |
|  | +D-GDA (ours) | **88.7** | **80.2** | **89.5** |

### B.2.4 Improvements over GSAGE backbone

We evaluate the performance of D-GDA using GSAGE (Graph SAmple and aggreGatE) [16] backbone using three benchmark datasets, including Cora, Citeseer, and Pubmed. In Table 22, we compare our results with existing SOTA methods on GSAGE backbone. D-GDA consistently improves over both baseline GSAGE and SOTA methods. Specifically, D-GDA outperforms the GSAGE baseline by 7.4%, 9.6%, and 7.4%, and exceeds the previous SOTA performance by 5.3%, 7.5%, and 7.4% on Cora, Citeseer, and Pubmed, respectively.

### B.3 Implementation Details for Link Prediction

To enhance data augmentation for link prediction tasks, for TSS, we train 2-layer Graph Convolutional Network (GCN) with a hidden dimension of 32 to generate node embeddings, followed by a simple

Table 23: (Table 7 extension) Balanced accuracy (%) comparison under class imbalance ratio of $\rho = 100$.

| Method | Cora-LT | Citeseer-LT |
|---|---|---|
| GCN (Vanilla) | 59.42±0.74 | 44.64±0.42 |
| +Reweight | 78.42±0.10 | 63.61±0.22 |
| +cRT | 76.54±0.22 | 60.60±0.25 |
| +PC Softmax | 77.30±0.13 | 62.15±0.45 |
| +CB Loss | 77.97±0.19 | 61.47±0.51 |
| +Focal Loss | 78.43±0.19 | 59.66±0.38 |
| +ReNode [8] | 67.61±0.13 | 47.78±0.31 |
| +Upsample | 75.52±0.11 | 55.05±0.11 |
| +DR-GCN | 73.90±0.29 | 56.18±1.10 |
| +GraphSMOTE [87] | 66.29±0.43 | 44.40±0.29 |
| +GraphENS [49] | 70.31±0.24 | 55.42±0.35 |
| +TAM (G-ENS) [59] | 72.10±0.23 | 57.15±0.34 |
| +GraphSHA [37] | 74.62±0.29 | 59.04±0.41 |
| +GraphVCM [52] | 75.81±0.42 | 60.53±1.37 |
| +D-GDA (ours) | **80.34±0.51** | **64.95±0.18** |

dot product to predict links. It is trained using the Adam optimizer with a learning rate of 0.001 for 1000 epochs, incorporating early stopping with a patience of 20 epochs to prevent overfitting. For all link prediction datasets, we set augmentation budget in the Target sample selector (TSS) to 20% of total number of training links for all datasets. The GVAE and LDM for link prediction are trained with the same parameters as those used for node classification.

### B.4 Implementation Details for Graph Classification

To enhance data augmentation for graph classification tasks, we begin by training a Target Sample Selector (TSS) to identify the most informative graphs for augmentation. The TSS employs a 2-layer GCN with a hidden dimension of 32 to compute node embeddings, which are subsequently aggregated into graph-level representations using a mean readout layer. The model is optimized using the Adam optimizer with a learning rate of 0.001 and trained for up to 1000 epochs, employing early stopping with a patience of 20 epochs to mitigate overfitting. For all graph classification datasets, we set augmentation budget in the Target sample selector (TSS) to 20% of total number of training graphs for all datasets. This targeted approach ensures augmentation focuses on samples with the highest potential impact. The GVAE and LDM used for augmentation are trained with the same hyperparameters as those employed in the node classification setting.

### B.5 Extended Experiments for Class Imbalance Handling Methods

The TSS, GVAE, and LDM components follow the same training procedures as described in the node classification setup (Section B.2).

#### B.5.1 Extended comparison with SOTA Methods

Table 23 provides an extended comparison of D-GDA with existing SOTA methods for handling class imbalance in the node classification task. This table serves as an extension of Table 7 presented in the main paper.

#### B.5.2 Imbalance Ratio Analysis

The Imbalance Ratio (IR) is defined as the ratio of the sample size of the largest majority class to that of the smallest minority class [10], with higher IR values indicating more severe class imbalance. Many existing graph imbalance handling methods attempt to mitigate this by generating synthetic nodes for all classes to match the majority class [87, 37, 49], thereby balancing the training data. However, such approaches often introduce unnecessary complexity and may lead to overfitting. In contrast, D-GDA effectively reduces class imbalance without enforcing class-level symmetry or generating synthetic nodes across all classes. At the core of D-GDA is the Targeted Sample

Selector (TSS), which identifies the nodes, regardless of their class, that are most likely to benefit from augmentation. This class-agnostic selection enables a more natural and targeted correction of imbalance. As a result, D-GDA leads to a significant reduction in the imbalance ratio. For instance, the IR drops from 113.67 to 19.55 on Cora, and from 123.67 to 14.6 on Citeseer, as shown in Table 24.

Table 24: Comparison of the augmentation budget (AugBudget) for reducing class imbalance ratio (IR) using D-GDA and SOTA methods.

| Datasets | Methods | #Training Nodes per Class | IR | AugBudget |
|----------|---------|---------------------------|-----|-----------|
| **Cora** | Original Data | [34, 7, 158, 341, 73, 15, 3] | 113.67 | - |
| | SOTA (Table 22) Augmentation | [341, 341, 341, 341, 341, 341, 341] | 1 | 278% |
| | D-GDA Augmented Data | [85, 25, 160, 391, 78, 39, 20] | 19.55 | 26.4% |
| **Citeseer** | Original Data | [3, 23, 371, 147, 58, 9] | 123.67 | - |
| | SOTA (Table 22) Augmentation | [371, 371, 371, 371, 371, 371] | 1 | 264% |
| | D-GDA Augmented Data | [30, 66, 438, 208, 66, 36] | 14.6 | 38.13% |

## B.6   D-GDA Improvements in ML Safety Measures (More Details)

We evaluate D-GDA across three key machine learning (ML) safety measures, including calibration, corruption robustness, and consistency, using three widely adopted benchmark datasets: Cora, Citeseer, and Pubmed. Table 8 in the main paper summarizes the results. All models are trained on clean versions of the datasets, i.e., the original data for the GCN baseline, and augmented versions for D-GDA and other GDA baselines. Performance is then assessed across the aforementioned safety tasks. Below, we define each safety measure, describe its objective, and provide details on the corresponding evaluation metrics.

**Calibration:** The goal of the calibration task is to classify nodes with calibrated prediction probabilities, i.e. matching the empirical frequency of correctness. In other words, a model that predicts with 80% confidence should be correct approximately 80% of the time. To evaluate this, we use the Root Mean Squared (RMS) Calibration Error [20] defined as: $\sqrt{\mathbb{E}_c[(\mathcal{P}(Y = \hat{Y}|C = c) - c)^2]}$, where $C$ is the classifier confidence that it's prediction $\hat{Y}$ is correct. We compute the RMS calibration error using 15 bins to estimate the empirical accuracy at different confidence levels.

**Corruption:** The objective of this task is to evaluate model robustness by classifying corrupted test nodes, with performance measured using the mean classification error. We assess D-GDA under four types of feature corruption: Gaussian noise, shot noise, impulse noise, and feature shift. Following the protocol from [19], we apply five severity levels to the Gaussian (G), shot (S), and impulse (I) noise corruptions to simulate varying degrees of degradation. For the feature shift corruption, we randomly select 10% of the test nodes and replace their features with those of a randomly chosen one-hop neighbor. It is important to emphasize that only test node features are corrupted, and the model is never trained on corrupted data. Below, we include the Python code used to generate these corrupted node features.

```python
def gaussian_noise(feature, severity=1):
    c = [0.04, 0.06, .08, .09, .10][severity - 1]
    return feature + np.random.normal(size=feature.shape, scale=c)

def shot_noise(feature, severity=1):
    c = [500, 250, 100, 75, 50][severity - 1]
    return np.random.poisson(feature * c)

def impulse_noise(feature, severity=1):
    c = [.01, .02, .03, .05, .07][severity - 1]
    return sk.util.random_noise(feature, mode='s&p', amount=c)

def shift(node, features, adj):
```

```
14        one_order_nei = adj[node].nonzero()[1]
15        neighbors = np.append(one_order_nei, node)
16        swap_with = random.choice(neighbors)
17        corr_feat = features.clone()
18        corr_feat[node], corr_feat[swap_with] = corr_feat[swap_with], corr_feat[node]
19        return corr_feat
```

**Consistency:** This task evaluates a model's ability to consistently classify similar versions of the same test node. For each node, we create a sequence $S$ containing the original node's features and several perturbed versions with increasing noise levels. The goal is for the model to predict the same class for all nodes within a sequence $S$, even as the features become progressively noisier. We denote $m$ perturbation sequences with $S = \{x_1^{(i)}, x_2^{(i)}, ..., x_n^{(i)}\}$. Following [19], we set $n = 31$. To measure consistency, we use the mean flip rate (mFR), as defined in Eq. (2). The mFR corresponds to the probability that adjacent nodes within a sequence $S$ of increasing noise levels have different predicted classes.

$$\text{mFR} = \frac{1}{m(n-1)} \Sigma_{i=1}^m \Sigma_{j=2}^n \mathbb{1}(f(x_j^{(i)}) \neq f(x_1^{(i)})), \tag{2}$$

where, $f(\cdot)$ is the trained classifier and $x_1$ is the original unperturbed features. Below is the Python code to obtain sequence $S$ of original features.

```
1   def get_sequence(features, n=31):
2       seq = []
3       for i in range(1, n+1):
4           features = features + 0.02 * torch.randn_like(features)
5           seq.append(features)
6       return seq
```

### B.7    D-GDA Adversarial Robustness Analysis (More Details)

To evaluate the robustness of D-GDA against adversarial attacks, we employ four well-established evasion attacks: Random (random edge flips), DICE [74] (removes intra-class edges and adds inter-class edges), GFAttack [6] (optimizes a low-rank loss to generate structural perturbations), and Meta-attack [93] (uses meta-gradient-based loss maximization). Each attack is applied at two perturbation levels, with perturbation ratios ($\sigma \in 0.05, 0.2$). All attacks are conducted under an evasion attack setting, where the graph structure is perturbed only at test time, leaving the training data unchanged. For GCN, the model is trained on clean data, whereas D-GDA is trained using its graph-augmented data. We leverage the GreatX: Graph Reliability Toolbox [33] to generate the adversarially perturbed graphs. Table 9 in the main paper summarizes the results.

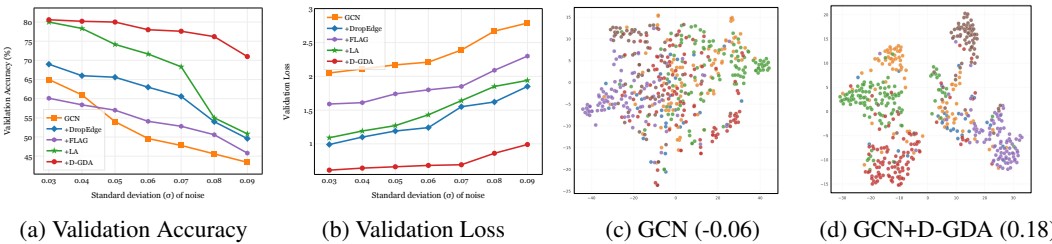

|  (a) Validation Accuracy  |  (b) Validation Loss  |  (c) GCN (-0.06)  |  (d) GCN+D-GDA (0.18)  |

Figure 5: **Citeseer dataset:** (a) and (b): D-GDA promotes flatter minima in the loss landscape. (c) and (d): t-SNE visualizations of node embeddings, demonstrating improved class separability with D-GDA. Silhouette score is shown in parenthesis.

## B.8 D-GDA Robustness to Homophily

The node homophily is defined as the average fraction of a node's neighbors that share the same label [50]. Table 25 compares these node homophily scores with GCN and D-GDA performance, as well as the resulting performance gain across six node classification and three link prediction datasets. Pearson's correlation coefficient between node homophily and these performances is computed. We observe a positive correlation of 0.63 between homophily and GCN performance, however, a reduced correlation of 0.45 with D-GDA performance. It suggests that D-GDA performance is not strongly correlated with homophily compared to GCN. More interestingly, we observe a negative correlation of -0.37 between homophily and performance gain of D-GDA, indicating that D-GDA performance gain is robust to homophily. On the remaining datasets, node labels are not available, so homophily can not be computed.

Table 25: Node Homophily Vs GCN and D-GDA performance: The D-GDA performance gain is robust to homophily.

| Downstream Task | Datasets | Node Homophily | GCN | D-GDA | Performance Gain |
|---|---|---|---|---|---|
| Node Classification | Cora | 0.83 | 81.6 | 89.1 | 9.19 |
| | Flickr | 0.67 | 61.2 | 87.3 | 42.65 |
| | Citeseer | 0.72 | 71.6 | 81.5 | 13.83 |
| | Pubmed | 0.79 | 78.8 | 88.2 | 11.93 |
| | Ogbn-Arxiv | 0.63 | 71.62 | 74.8 | 4.44 |
| | Ogbn-Products | 0.83 | 71.37 | 78.85 | 10.48 |
| Link Prediction | Cora | 0.83 | 89.55 | 96.96 | 8.27 |
| | Citeseer | 0.72 | 69.47 | 93.72 | 34.91 |
| | Pubmed | 0.79 | 96.11 | 97.82 | 1.78 |
| **Pearson's Correlation** | | - | 0.63 | 0.45 | -0.37 |

## B.9 More insights to the improvements obtained by D-GDA

### B.9.1 D-GDA promotes flatter minima for enhanced generalization

In addition to the Cora dataset results in the main paper, we evaluate D-GDA on the Citeseer and Pubmed datasets. Using a trained GCN, we compare D-GDA against DropEdge, FLAG, and LA under Gaussian noise perturbations of model parameters. As shown in Figure 5(a), D-GDA achieves higher accuracy and lower loss (Figure 5(b)) on Citeseer, indicating flatter minima that enhance generalization. t-SNE visualizations (Figure 5(c) and (d)) reveal more cohesive and well-separated clusters for D-GDA, with the silhouette score improving from -0.06 to 0.18. A similar trend is observed on Pubmed (Figure 6), where the silhouette score increases from 0.13 to 0.34, further confirming D-GDA's ability to promote robust and generalizable representations.

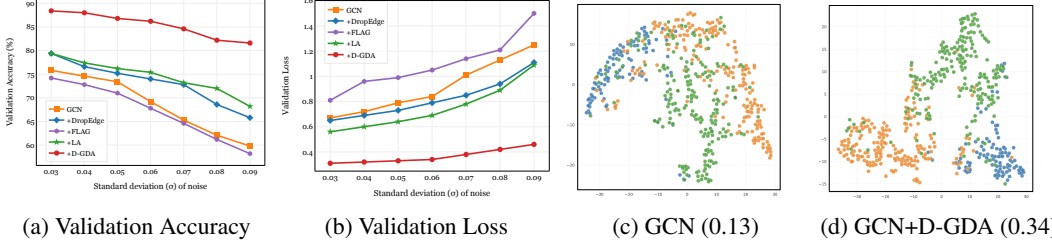

(a) Validation Accuracy     (b) Validation Loss     (c) GCN (0.13)     (d) GCN+D-GDA (0.34)

Figure 6: **Pubmed dataset:** (a) and (b): D-GDA promotes flatter minima in the loss landscape. (c) and (d): t-SNE visualizations of node embeddings, demonstrating improved class separability with D-GDA. Silhouette score is shown in parenthesis.

### B.9.2 Oversmoothness analysis using MADGap measure (More Details)

To quantify the smoothness of graph representations, we employ the MADGap measure, as described by Chen et al. [7]. The results are presented in Table 10 of the main paper. Below, we outline the

MADGap measure and its computation. The Mean Average Distance (MAD) assesses graph representation smoothness by computing the average distance between nodes. Given a graph representation matrix $\boldsymbol{H} \in \mathbb{R}^{n \times d'}$ (where $d'$ is the hidden dimension of the final GNN layer), we first calculate the distance matrix $D \in \mathbb{R}^{n \times n}$ using the cosine distance between all node pairs. We then filter target node pairs by element-wise multiplication with a mask matrix $M^{\text{tgt}}$ to obtain $D^{\text{tgt}} = D \odot M^{\text{tgt}}$. The average distance for each row in $D^{\text{tgt}}$ is computed as:

$$\bar{D}^{tgt} = \frac{\Sigma_{j=0}^n D_{ij}^{tgt}}{\Sigma_{j=0}^n \mathbb{1}(D_{ij}^{tgt})} \tag{3}$$

where $\mathbb{1}(\cdot)$ is the indicator function. The MAD score for the target node pairs is then calculated by averaging the non-zero $\bar{D}_i^{\text{tgt}}$ values:

$$MAD^{tgt} = \frac{\Sigma_{j=0}^n \bar{D}_{ij}^{tgt}}{\Sigma_{j=0}^n \mathbb{1}(\bar{D}_{ij}^{tgt})} \tag{4}$$

Using the graph topology to approximate node categories, we compute the MADGap to evaluate oversmoothness:

$$\text{MADGap} = \text{MAD}^{\text{rmt}} - \text{MAD}^{\text{neb}}, \tag{5}$$

where $\text{MAD}^{\text{rmt}}$ is the MAD for remote nodes (order $\geq 8$) and $\text{MAD}^{\text{neb}}$ is the MAD for neighboring nodes (order $\leq 3$). A large positive MADGap indicates that nodes receive more useful information than noise, reflecting appropriate smoothing and good GNN performance. Conversely, a small or negative MADGap suggests oversmoothing, leading to degraded performance.

### B.9.3 Test-Time Consistency and Diversity (More Details)

To evaluate the quality of augmented data, we adopt the consistency and diversity metrics proposed by Bo et al. [3] and extend them to the test set. We train two models, $\mathcal{F}_\theta$ and $\tilde{\mathcal{F}}_\theta : \mathbb{R}^d \to \mathbb{R}^c$, on the original training data $D_{\text{train}}$ and augmented data $\tilde{D}_{\text{train}}$, respectively, where $d$ is the input feature dimension, $c$ is the number of classes, and $\theta$ represents the learnable parameters. Both models are then used to predict on the test set $D_{\text{test}}$. Effective augmentations should enable $\tilde{\mathcal{F}}_\theta$ to achieve higher test accuracy and establish a distinct decision boundary compared to $\mathcal{F}_\theta$.

**Test-Time Consistency**: We measure consistency as the accuracy of the augmented model on the test set, defined as $C_{\text{test}} = \text{Acc}(\tilde{\mathcal{F}}_\theta(D_{\text{test}}), Y_{\text{test}})$, where $Y_{\text{test}}$ denotes the test data labels. A low $C_{\text{test}}$ indicates that the augmentations are inconsistent with the original data, potentially reducing model accuracy. However, a high $C_{\text{test}}$ does not necessarily imply high-quality augmentations, as it may reflect limited generalization.

**Test-Time Diversity**: To assess diversity, we compute the difference between the predictions of the original and augmented models using the Frobenius norm: $D_{\text{test}} = \|\tilde{\mathcal{F}}_\theta(D_{\text{test}}) - \mathcal{F}_\theta(D_{\text{test}})\|_F^2$. A low $D_{\text{test}}$ suggests that the augmented data closely resembles the original data, offering little benefit to model generalization (Yin et al., 2019). Conversely, a high $D_{\text{test}}$ does not guarantee correct augmentations, as it may introduce noise.

A robust augmentation strategy should balance high consistency ($C_{\text{test}}$) and sufficient diversity ($D_{\text{test}}$) to ensure both accuracy and improved generalization. Figure 7 illustrates the trade-off between diversity and consistency for D-GDA compared to existing SOTA augmentation methods. Across all datasets, D-GDA achieves a favorable balance between the two, suggesting that its performance gains may stem from this effective trade-off.

### B.10 Additional Ablations

In this section, we present additional ablation studies beyond those in the main paper to provide a clearer understanding of various design choices.

### B.10.1 Impact of Neighborhood Aggregation Method on Performance

To assess the impact of different neighborhood aggregation strategies, we experiment with two commonly used methods: mean and max aggregation. These strategies are used to compute the conditioning vector $\boldsymbol{c}_i^0$ for augmentation generation. As shown in Table 26, the mean aggregation yields

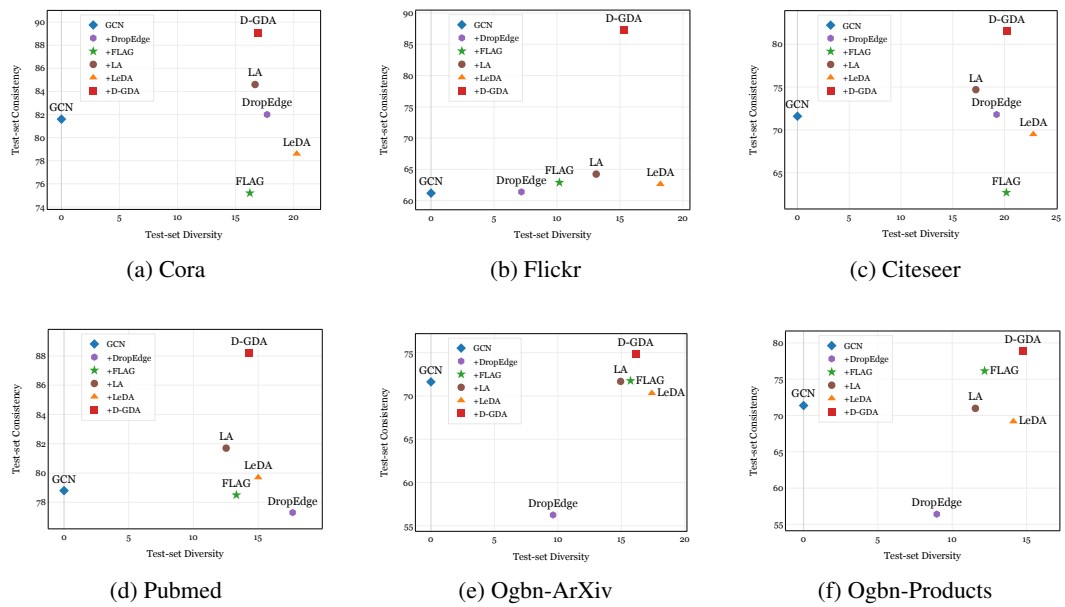

(a) Cora       (b) Flickr       (c) Citeseer

(d) Pubmed       (e) Ogbn-ArXiv       (f) Ogbn-Products

Figure 7: (Details of Figure 1b) Diversity vs. consistency comparison on six datasets. Methods falling in top-right corner shows a good balance of diversity vs. consistency

superior performance, indicating its effectiveness in capturing relevant neighborhood information for guiding the augmentation process.

Table 26: Impact of Neighborhood Aggregation Method on Performance

| Aggregation Method | Cora | Citeseer | Pubmed |
|---|---|---|---|
| Max | 87.5 | 78.8 | 84.7 |
| Mean (ours) | **89.1** | **81.5** | **88.2** |

### B.10.2   Impact of Multi-Hop Neighborhood Condition on Performance

To assess the effect of multi-hop neighborhood condition on model performance, we conducted an ablation study by modifying the conditioning vector $c_i^0$ in the LDM to incorporate embeddings from 1-hop, 2-hop, and 3-hop neighboring nodes. The results, summarized in Table 27, show a decline in performance as neighborhood depth increases. This may be attributed to the changing class labels of nodes in deeper neighborhoods, which could prevent the synthetic node features from accurately capturing the target class's characteristics. Notably, as depicted in Figure 8, increasing the neighborhood depth enhances test-time diversity but reduces test-time consistency.

Table 27: Impact of Multi-hop Neighborhood Aggregation on overall performance.

| Conditioned On | Cora | Citeseer | Pubmed |
|---|---|---|---|
| 1-hop neighbors (ours) | **89.1** | **81.5** | **88.2** |
| 2-hop neighbors | 88.2 | 79.9 | 87.9 |
| 3-hop neighbors | 86.8 | 77.6 | 85.3 |

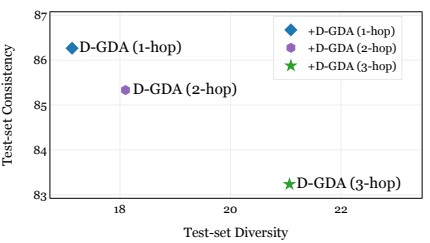

Figure 8: Multi-hop Neighborhood impact on test-time consistency and diversity

### B.10.3 Node Importance Selection Strategies for Graph Classification

To evaluate the effectiveness of the proposed degree-based node selection method, we compared it against two alternative strategies including random selection and PageRank-based selection. Table 28, summarizes the results for graph classification across these strategies. Following [15], we compute PageRank with a damping factor of 0.85 and a convergence tolerance of $1 \times 10^{-6}$. We observe that the proposed degree-based selection consistently outperforms both alternatives. This supports the intuition that high-degree nodes are indeed more structurally influential, reinforcing the effectiveness of degree-based selection as the optimal strategy in our setting.

Table 28: D-GDA performance comparison on graph classification datasets using the proposed degree-based node selection strategy, compared against random and PageRank-based selection methods.

| Node Selection Strategy | ogbg-molSIDER | ogbg-molClinTox | ogbg-molHIV | ogbg-molBACE |
|---|---|---|---|---|
| Random node | 61.19 | 89.98 | 77.42 | 75.68 |
| Page-rank top-node | 63.24 | 92.76 | 78.83 | 81.92 |
| Highest degree node (ours) | **64.85** | **94.82** | **79.03** | **83.45** |

### B.10.4 Effectiveness of Test Time Augmentation (Extended Evaluation)

We present an extended evaluation of Test Time Augmentation (TTA) by expanding the analysis to 10 datasets to provide a more comprehensive view. Table 29, summarizes the results for GCN and D-GDA, both with and without TTA, along with the per-sample TTA cost. Across all datasets and tasks, we observe consistent performance improvements when employing TTA, highlighting its generalizability and effectiveness.

Table 29: Performance comparison of GCN and D-GDA with and without test time augmentation (TTA) along with Per-sample inference time (sec) with TTA

| Datasets | GCN | | D-GDA | | TTA Cost/Sample | Inference Time/Sample |
|---|---|---|---|---|---|---|
| | w/o TTA | with TTA | w/o TTA | with TTA | | |
| Cora | 81.6 | 83.2 | 88.2 | 89.1 | 1.39 | 1.41 |
| Flickr | 61.2 | 64.01 | 84.65 | 87.3 | 1.17 | 1.33 |
| Citeseer | 71.6 | 73.37 | 79.2 | 81.5 | 1.53 | 1.55 |
| Pubmed | 78.8 | 80.41 | 85.5 | 88.2 | 1.21 | 1.23 |
| Ogbn-Arxiv | 71.62 | 72.52 | 72.7 | 74.8 | 0.98 | 1.8 |
| Ogbn-Products | 71.37 | 72.85 | 77.60 | 78.85 | 1.02 | 1.94 |
| Ogbg-molSIDER | 59.6 | 61.08 | 62.92 | 64.85 | 0.96 | 1.08 |
| Ogbg-molClinTox | 88.55 | 89.92 | 91.8 | 94.82 | 0.97 | 1.11 |
| Ogbg-molHIV | 76.06 | 77.63 | 77.9 | 79.03 | 0.96 | 1.09 |
| Ogbg-molBACE | 71.47 | 73.26 | 79.24 | 83.45 | 0.95 | 1.06 |

### B.10.5 Ablation of Different Choices of Target Sample Selection (TSS) Backbones

In D-GDA, TSS identifies difficult samples based on prediction entropy. To maintain consistency, we use the same backbone architecture in TSS as in the downstream model being evaluated. For example, when comparing D-GDA performance with a GCN backbone, we also use GCN within TSS; similarly, for GAT or GraphSAGE, the corresponding backbone is used in both components. The rationale behind this design is that each GNN backbone employs a different message passing mechanism, and as a result, a node that may appear difficult for one model may not be difficult for another architecture. Table 30 summarizes the results of using GCN, GAT, and GraphSAGE within the TSS module across multiple experiments. The findings show that the best performance is consistently achieved when the same backbone is used in both TSS and the downstream classifier, confirming the importance of the proposed choice of architecture in TSS.

### B.10.6 Hyperparameter Analysis

To evaluate the influence of hyperparameters on the overall performance of D-GDA, we conduct an ablation study focusing on the weighting parameters $\lambda_1$ and $\lambda_2$ in the GVAE loss function using Cora, Citeseer, and Pubmed datasets for the node classification task. The parameters $\lambda_1$ and $\lambda_2$

Table 30: Ablation on different choices of TSS backbones.

| Downstream Backbone | TSS Backbone | Cora | Citeseer | Pubmed |
|---|---|---|---|---|
| GCN | GCN | **89.1** | **81.5** | **88.2** |
| | GAT | 87.8 | 79.8 | 85.5 |
| | GSAGE | 87.4 | 80.2 | 85.4 |
| GAT | GCN | 86.2 | 76.2 | 87.1 |
| | GAT | **89.4** | **79.0** | **88.9** |
| | GSAGE | 87.3 | 78.4 | 86.8 |
| GSAGE | GCN | 86.7 | 77.5 | 85.8 |
| | GAT | 87.3 | 78.9 | 86.1 |
| | GSAGE | **88.7** | **80.2** | **89.5** |

regulate the trade-off between feature reconstruction and latent space regularization, both of which are critical for generating high-quality augmented graph data. In this ablation, we systematically vary $\lambda_1 \in \{0.1, 0.3, 0.5\}$ and $\lambda_2 \in \{0.01, 0.05, 0.1\}$, while keeping all other components of D-GDA unchanged. For each pair of $\lambda_1$ and $\lambda_2$ values, we train the GVAE to generate augmented graphs, which are subsequently used to train the node classification model. Table 31 presents the results, reporting classification performance across different hyperparameter configurations. The analysis highlights the sensitivity of GVAE to the balance between feature reconstruction and regularization. For example, increasing $\lambda_1$ emphasizes feature reconstruction, which can improve node-level fidelity but may lead to overfitting. Conversely, higher $\lambda_2$ values impose stronger regularization on the latent space, promoting stability but potentially reducing the model's capacity to capture nuanced structural information.

Table 31: Hyperparameter Analysis

| $\lambda_1$ | $\lambda_2$ | Cora | Citeseer | Pubmed |
|---|---|---|---|---|
| 0.1 | 0.01 | 88.38 | 79.92 | 86.45 |
| | 0.05 | 85.5 | 75.36 | 84.68 |
| | 0.1 | 82.28 | 72.83 | 81.73 |
| 0.3 (ours) | 0.01 (ours) | **89.1** | **81.5** | **88.2** |
| | 0.05 | 87.12 | 79.8 | 87.02 |
| | 0.1 | 82.52 | 73.86 | 84.7 |
| 0.5 | 0.01 | 88.56 | 80.13 | 87.14 |
| | 0.05 | 85.23 | 78.25 | 84.88 |
| | 0.1 | 82.48 | 74.6 | 81.92 |

## B.11 D-GDA Training Time Breakdown

Table 32, provides a component-wise breakdown of the D-GDA training time, covering Traget Sample Selector (TSS), Graph Variational Autoencoder (GVAE), and Latent Diffusion Model (LDM), along with its inference time with Test-Time Augmentation (TTA). Training and inference time are reported using a 4x A16 GPU machine with 128GB RAM.

## B.12 D-GDA Scalability and Efficiency Analysis

Table 33, compares the scalability of D-GDA across datasets of varying numbers of nodes and number of graphs. Specifically, performance gain, training time, and per-sample inference time of D-GDA are compared with the dataset size. For datasets containing multiple graphs, we report the average number of nodes per graph. The results demonstrate that D-GDA scales efficiently with increasing dataset size. The training time increases sublinearly as the dataset size increases, while still delivering significant performance improvements over GCN.

## B.13 Limitations of the Proposed D-GDA Algorithm

A potential limitation of the proposed D-GDA framework is its training time computational cost and test-time memory overhead. It requires training GCN baseline, GVAE, and LDM for each dataset.

Table 32: Training (sec) and Inference time per sample (sec) component-wise breakdown of the proposed D-GDA framework.

| Datasets | Train Time | | | | Inference Time | | |
|---|---|---|---|---|---|---|---|
| | TSS | GVAE+LDM | Data Aug+Training | Total | TTA | Inference | Total |
| Cora | 9.35 | 12.82 | 15.55 | 37.72 | 1.39 | 0.019 | 1.41 |
| Flickr | 79.87 | 85.13 | 163.17 | 328.17 | 1.17 | 0.16 | 1.33 |
| Citeseer | 11.67 | 17.86 | 18.32 | 47.85 | 1.53 | 0.021 | 1.55 |
| Pubmed | 40.01 | 43.62 | 145.79 | 229.42 | 1.21 | 0.024 | 1.23 |
| Ogbn-Arxiv | 728.86 | 774.92 | 903.42 | 2407.2 | 0.98 | 0.816 | 1.8 |
| Ogbn-Products | 1253.84 | 1548.24 | 1698.23 | 4500.31 | 1.02 | 0.92 | 1.94 |
| ogbg-molSIDER | 95.11 | 105.32 | 113.72 | 314.15 | 0.96 | 0.12 | 1.08 |
| ogbg-molClinTox | 96.32 | 106.46 | 116.24 | 319.02 | 0.97 | 0.14 | 1.11 |
| ogbg-molHIV | 572.28 | 585.18 | 596.48 | 1753.94 | 0.96 | 0.13 | 1.09 |
| ogbg-molBACE | 96.84 | 103.92 | 114.62 | 315.38 | 0.95 | 0.114 | 1.06 |

Table 33: Scalability analysis of D-GDA with varying graph sizes (number of nodes |V| and number of graphs |G|), showing performance gains over the baseline GCN, along with total training and inference times (in seconds). D-GDA exhibits sublinear growth in training time with increasing dataset size.

| Datasets | |V| | |G| | Gain | Inf. Time | Train Time | Train Time Per Sample |
|---|---|---|---|---|---|---|
| Cora | 2,708 | 1 | 9.19 | 1.41 | 37.72 | 0.0139 |
| Flickr | 7,575 | 1 | 42.65 | 1.33 | 328.17 | 0.0433 |
| Citeseer | 3327 | 1 | 13.83 | 1.55 | 47.85 | 0.0144 |
| Pubmed | 19,717 | 1 | 11.93 | 1.23 | 229.42 | 0.0116 |
| Ogbn-Arxiv | 169,343 | 1 | 4.44 | 1.8 | 2407.2 | 0.0142 |
| Ogbn-Products | 2,449,029 | 1 | 10.48 | 1.94 | 4500.31 | 0.0018 |
| ogbg-molSIDER | 33.6 | 1,427 | 8.81 | 1.08 | 314.15 | 0.2201 |
| ogbg-molClinTox | 26.2 | 1,477 | 7.08 | 1.11 | 319.02 | 0.216 |
| ogbg-molHIV | 25.5 | 41,127 | 3.9 | 1.09 | 1753.94 | 0.0426 |
| ogbg-molBACE | 34.1 | 1,513 | 16.76 | 1.06 | 315.38 | 0.2084 |

This process is resource-intensive than some traditional augmentation methods, such as DropEdge. However, some traditional methods such as FLAG have also shown similar computational time. Being diffusion based framework, D-GDA requires GVAE and LDM during inference if test-time augmentation is employed. Without test-time augmentation its inference time remains almost the same as existing SOTA methods. Despite the higher computational overhead, the substantial performance improvements over SOTA methods justify the additional cost, making D-GDA a compelling choice for high-quality graph data augmentation.

## C   Convergence Proof of LDM in D-GDA Framework

We provide a theoretical justification for the convergence of LDM in the D-GDA framework setting, which operates in label-free settings. Despite the inherent challenges of sparse graph supervision, we show that the denoising network in D-GDA converges to the true conditional score function, enabling it to sample augmentations from a neighborhood-aware latent distribution. Our analysis follows the foundational results from score-based generative modeling [21, 62, 61], adapted to the graph setting with local structural conditioning. Let $v_i$ denote a node in graph $\mathcal{G} = (\mathcal{V}, \mathcal{E})$, with latent representation $z_i^0 \in (R)^d$ obtained from a Graph Variational Autoencoder (GVAE). We assume a standard DDPM-style forward process that gradually corrupts $z_i^0$ with Gaussian noise. The reverse process aims to estimate the noise via a denoising network $\epsilon_\theta(z_i^t, t, c_i^0)$, conditioned on a neighborhood-aware vector $c_i^0$. The model is trained with a score-matching loss: $\mathcal{L}_{LDM} = E_{z_i^0, t, \epsilon} ||\epsilon - \epsilon_\theta(z_i^t, t, c_i^0)||_2^2.$. For details See Section 3.1.3. Our goal is to show that under mild assumptions, the denoising model $\epsilon_\theta$ converges to the true conditional score function of the local latent distribution, enabling realistic and diverse augmentations without reliance on class labels. We begin by stating the following assumptions:

**Assumption 1 (Locality Assumption):** The latent variable $z_i^0$ is drawn from a smooth conditional distribution $p(z_i^0 \mid \mathcal{N}(v_i))$, supported on a low-dimensional manifold.

Table 34: Impact of reducing training data for LDM on training loss, validation loss, early stopping epoch, and node classification accuracy.

| Training Data Percentage | Performance Metric | Cora | Citeseer | Pubmed | Ogbn-Arxiv |
|---|---|---|---|---|---|
| 100% | Train Loss | 0.069 | 0.084 | 0.077 | 0.068 |
| | Val Loss | 0.153 | 0.176 | 0.249 | 0.251 |
| | Test Acc. | 89.1 | 81.5 | 88.2 | 74.8 |
| 60% | Train Loss | 0.059 | 0.064 | 0.067 | 0.057 |
| | Val Loss | 0.178 | 0.197 | 0.275 | 0.307 |
| | Test Acc. | 86.8 | 79.2 | 86.4 | 73.1 |
| 40% | Train Loss | 0.029 | 0.034 | 0.042 | 0.032 |
| | Val Loss | 0.203 | 0.236 | 0.319 | 0.324 |
| | Test Acc. | 82.3 | 75.6 | 81.9 | 71.8 |

**Assumption 2 (Model Capacity):** The denoising network $\epsilon_\theta$ is a universal approximator i.e. has enough expressive power to learn any function within a given function class (For D-GDA the score function of the noisy latent distribution).

**Assumption 3 (Diffusion Schedule):** The noise schedule $\{\alpha_t\}_{t=1}^{T}$ is chosen such that $\bar{\alpha}_T \to 0$ as $T \to \infty$, ensuring near-total corruption at the final step.

**Theorem 1:** Let $\epsilon_\theta(z_i^t, t, c_i^0)$ be trained using the denoising score matching objective. Then, as $T \to \infty$ and model capacity increases, $\epsilon_\theta$ converges to the conditional score function: $\epsilon_\theta(z_i^t, t, c_i^0) \to \nabla_{z_i^t} \log p_t(z_i^t \mid c_i^0)$, where $p_t(z_i^t \mid c_i^0)$ is the marginal distribution of the noisy latent variable at time $t$, conditioned on the local structure of node $v_i$.

**Sketch of Proof:** This result follows from established results in score-based generative modeling [61]. Minimizing the denoising score-matching loss recovers the true score function of the noisy data distribution. In D-GDA, the conditioning vector $c_i^0$ restricts generation to a node's local neighborhood, simplifying the modeling of $p(z_i^0 \mid N(v_i))$ as a smooth, low-dimensional conditional distribution. Assumption A1 ensures the score function is well-defined and smooth, while Assumption A2 guarantees that $\epsilon_\theta$ can approximate this function arbitrarily well. The noise schedule in A3 ensures that the diffusion process sufficiently corrupts the latent code, enabling meaningful denoising learning. Together, these yield convergence to the true conditional score function. This theoretical justification confirms that D-GDA's latent diffusion process, when conditioned on local neighborhood structure, can provably learn a neighborhood-aware augmentation distribution, without requiring access to class labels.

**Empirical validation:** we evaluate D-GDA's LDM under increasingly limited training data. Specifically, we reduce the amount of training data from 100% to 60% and 40%, and track LDM behavior across four graph datasets: Cora, Citeseer, Pubmed, and Ogbn-Arxiv. Early stopping is applied based on validation loss with a patience of 10 epochs. In Table 34, we report the training loss, validation loss, early stopping epoch, and D-GDA test accuracy. We observe that, as the training data decreases, training loss consistently decreases (e.g., Cora: 0.069 to 0.029), due to easier overfitting on fewer samples. Validation loss increases (e.g., Pubmed: $0.249 \to 0.319$), indicating reduced generalization from limited training data. LDM trained with less data tend to converge earlier (e.g., Ogbn-Arxiv: $672 \to 172$), suggesting faster overfitting. Test accuracy drops across datasets (e.g., Citeseer: 81.5% $\to$ 75.6%), as the model struggles to generate diverse augmentations.

