# OpenReview forum: "Diffusion-Guided Graph Data Augmentation"
_NeurIPS.cc/2025/Conference — NeurIPS 2025 poster_

### Official Review · Reviewer_N1DJ · 2025-06-30

**Clarity:** 2
**Significance:** 3
**Originality:** 3
**Rating:** 5
**Confidence:** 3

**Summary:**

The paper proposes Diffusion-Guided Graph Data Augmentation (D-GDA), a novel framework to enhance the performance and generalization of Graph Neural Networks (GNNs) by generating diverse and consistent graph augmentations. The key components of D-GDA include Target Sample Selector (TSS), Graph Variational Autoencoder (GVAE) and Latent Diffusion Model (LDM).

The method is task-agnostic and label-free, enabling test-time augmentation. Extensive experiments on 12 benchmarks show superior performance over 30 SOTA methods, with improvements in generalization, robustness to adversarial attacks, and ML safety measures.

**Questions:**

See weakness.

**Ethical Concerns:**

["NO or VERY MINOR ethics concerns only"]

**Final Justification:**

Thank you for your detailed response. The author has addressed all of my concerns thoroughly, and I am pleased to revise my rating accordingly.

**Limitations:**

Yes.

**Paper Formatting Concerns:**

No.

**Quality:**

3

**Strengths And Weaknesses:**

Strengths:

1. The paper introduces a label-free, diffusion-based GDA framework applicable to multiple graph tasks, unlike prior task-specific methods.
2. Neighborhood-aware conditioning ensures augmentations preserve local structure while diversifying data.

Weaknesses:

1. The paper claims to balance diversity and consistency in augmentations but does not explicitly define how these metrics are measured. Are they based on feature variance or some other criterion?
2. While diversity and consistency are highlighted as key advantages, the paper does not explain why these metrics matter and whether they conflict (e.g., more diversity may reduce consistency).
3. The TSS relies on a baseline GNN to identify "difficult" samples via entropy. However, the choice of baseline architecture (e.g., GCN or GraphSAGE) may bias sample selection. Is this evaluated?
4. The paper claims lower computational costs due to latent-space diffusion but does not provide theoretical analysis or empirical benchmarks. Without these, the efficiency claims remain unsubstantiated.
5. The method is described as using a Graph Variational Autoencoder (GVAE), but the cited work refers to Variational Graph Autoencoders (VGAE). Is this a terminological inconsistency or a modified architecture?

---

> ### Author Rebuttal · Authors · 2025-07-31
>
> # 1) Definition and Measurement of Consistency and Diversity
> The definitions of consistency and diversity [3], along with how to measure them, have already been provided in Supplementary Section B.8.3; however, we will revise the main paper to explicitly reference this section for clarity. Specifically, **Consistency** is measured as the test accuracy of the model trained on augmented data, reflecting how well the augmentation preserves task-relevant semantics, and **Diversity** is measured as the Frobenius norm of the difference between the prediction logits of the original and augmented models, capturing how much variation the augmentation introduces in the model’s outputs. For more details, please refer to Supplementary Section B.8.3.
>
> # 2) Importance of Consistency and Diversity
> Consistency and diversity [3, R5, R6] are used as key indicators for measuring augmentation quality. Consistency ensures that augmentations preserve task-relevant semantics, leading to stable predictions. Diversity encourages variations that expose the model to different structural or feature-level perturbations, promoting better generalization. While these objectives can indeed conflict (i.e., excessive diversity may harm consistency), an effective augmentation strategy must balance both to avoid overfitting to trivial perturbations or being too invariant to informative changes. In our work, we demonstrate that existing methods tend to lean towards one of these at the expense of the other (Figure 1b). In contrast, D-GDA is able to strike a good balance between both.
>
> [R5] *Juan, Xin, et al. "Multi-strategy adaptive data augmentation for Graph Neural Networks." Expert Systems with Applications 258 (2024): 125076.*
>
> [R6] *Chen, Nan, et al. "Consistency training with learnable data augmentation for graph anomaly detection with limited supervision." The twelfth international conference on learning representations. 2024.*
>
> # 3) Ablation on different Choices of Target Sample Selection (TSS) Backbones
> In D-GDA, TSS identifies difficult samples based on prediction entropy. To maintain consistency, we use the same backbone architecture in TSS as in the downstream model being evaluated. For example, when comparing D-GDA performance with a GCN backbone, we also use GCN within TSS; similarly, for GAT or GraphSAGE, the corresponding backbone is used in both components. The rationale behind this design is that each GNN backbone employs a different message passing mechanism, and as a result, a node that may appear difficult for one model may not be difficult for another architecture. Table 9 summarizes the results of using GCN, GAT, and GraphSAGE within the TSS module across multiple experiments. The findings show that the best performance is consistently achieved when the same backbone is used in both TSS and the downstream classifier, confirming the importance of the proposed choice of architecture in TSS.
>
> **Table 9:** Ablation on different choices of TSS backbones.
>
> |Downstream Backbone| TSS Backbone |Cora| Citeseer|Pubmed|
> |-----------|-----------|-----------|-----------|-----------|
> |GCN|GCN|**89.1**|**81.5**|**88.2**|
> |GCN|GAT|87.8|79.8|85.5|
> |GCN|GSAGE|87.4|80.2|85.4|
> |GAT|GCN|86.2|76.2|87.1|
> |GAT|GAT|**89.4**|**79.0**|**88.9**|
> |GAT|GSAGE|87.3|78.4|86.8|
> |GSAGE|GCN|86.7|77.5|85.8|
> |GSAGE|GAT|87.3|78.9|86.1|
> |GSAGE|GSAGE|**88.7**|**80.2**|**89.5**|
>
> # 4) Diffusion Model and Latent Diffusion Model (LDM) Computational Cost Comparison
> Rombach et al. [R7] demonstrated that training a state-of-the-art diffusion model requires approximately 1000 V100 GPU-days, whereas training a Latent Diffusion Model (LDM) reduces this cost to just 271 V100 GPU-days, reflecting a 3.6× improvement in training time. This computational advantage of LDMs has been consistently highlighted in prior works such as [5, 20, 66]. We outline our rationale for adopting latent diffusion in Section 3.1.3 of the main paper. Specifically, Latent Diffusion Models (LDMs) offer a significantly more efficient and tractable alternative to applying diffusion directly in the raw feature space. Especially for graph data, training diffusion models in the raw feature space presents two key challenges: 1) **High-dimensional sparsity: ** Many graph datasets, such as Cora and Citeseer, contain sparse and high-dimensional feature vectors. Modeling directly in this space makes diffusion model training unstable and computationally expensive. 2) **Structural complexity:** Preserving both topological and semantic information during diffusion model training in the raw space demands more complex architectures and heavier training models, further increasing the resource burden.
>
> To overcome these issues, our method applies diffusion in a compact latent space learned via our Graph Variational Autoencoder (GVAE). This latent representation captures the essential structural and feature-level information while significantly reducing dimensionality and sparsity. As a result, latent diffusion not only lowers computational cost but also enhances training stability and scalability. We will revise the manuscript to more clearly communicate this motivation to further strengthen the theoretical grounding of our design choice. In Table 14 main paper, removal of LDM resulted in reduced performance on three datasets.
>
> [R7] *Rombach, Robin, et al. "High-resolution image synthesis with latent diffusion models." Proceedings of the IEEE/CVF conference on computer vision and pattern recognition. 2022.*
>
> # 5) Clarification between VGAE and GVAE
> We cited VGAE [29] as it introduced the foundational concept of applying variational autoencoders to graph-structured data. However, our use of the term GVAE (Graph Variational Autoencoder) reflects a generalized and modified architecture beyond what VGAE originally proposed. Specifically, VGAE was designed for link prediction and employs a dot-product decoder that only reconstructs edge presence. In contrast, our GVAE includes two distinct decoders: a feature decoder that reconstructs node features, and an edge decoder that reconstructs edge connectivity. Please see Section 3.1.2 for more details.

---

> > ### Comment · Reviewer_N1DJ · 2025-08-06
> >
> > Thank you for addressing the rebuttal. I find that most of my concerns have been satisfactorily resolved. However, I maintain one reservation:
> > In Section 3.1.2, the third sentence introduces GVAE, yet it cites a reference pertaining to VGAE. Could this indicate a discrepancy in citation accuracy?

---

> ### Author Response · Authors · 2025-08-06
> **Further Clarification between GVAE and VGAE**
>
> We thank the respected reviewer for encouraging comments. To incorporate the suggestion of the respected reviewer, the third sentence in Section 3.1.2, is modified as : “Variational graph autoencoders learn expressive latent representations of graph data, capturing both node features and local structure [28,33,45]. Therefore, we employ a variant of this architecture for compact latent representation learning, resulting in reduced computational complexity and improved performance. Our GVAE consists of a GCN-based encoder and two decoders for node feature and edge reconstruction.”

---

> > ### Comment · Reviewer_N1DJ · 2025-08-08
> > **Reply to Author**
> >
> > Thank you for your detailed response. The author has addressed all of my concerns thoroughly, and I am pleased to revise my rating accordingly.

---

> > > ### Author Response · Authors · 2025-08-08
> > >
> > > We sincerely appreciate your time and constructive feedback. We are pleased that we were able to address all of your concerns.

---

### Official Review · Reviewer_MRas · 2025-07-01

**Clarity:** 3
**Significance:** 2
**Originality:** 2
**Rating:** 4
**Confidence:** 3

**Summary:**

This paper proposes a Graph Data Augmentation (GDA) method called D-GDA, which is based on Graph Variational Autoencoder (GVAE) and Latent Diffusion Model (LDM). The framework consists of three key components: i). Target sample selection（TSS）, ii). Graph variational auto-encoder（GVAE）, and iii) Latent Diffusion.  In addition, D-GDA is model-agnostic and label-free, making it highly flexible for various applications. Extensive experiments demonstrate its effectiveness.

**Questions:**

Please refer to the weaknesses.

**Ethical Concerns:**

["NO or VERY MINOR ethics concerns only"]

**Final Justification:**

My concerns have been addressed , and I have increased both the Clarity and Final Rating scores to 3 and 4, respectively.

**Limitations:**

yes

**Quality:**

3

**Strengths And Weaknesses:**

Strengths:

1. The framework of D-GAD is sample and model-agnostic, enabling it enabling it to be applied more flexibly.

2. D-GDA not only focuses on performance improvement in the GDA scenario but also conducts research on model security.

3. The experiments compare performance across multiple tasks with numerous baseline models and ample benchmark datasets, providing comprehensive and thorough comparative analysis.

Weaknesses：

1. This paper lacks a structured theoretical description of TSS, GVAE, and LDM in the Methodology section(Sec. 3), which hinders a clear understanding of the technical approach.

2. GDA is inherently designed for scenarios with sparse or restricted graph data. However, LDM's convergence requires large amounts of data and significant computational resources. A theoretical proof of LDM's convergence in the current D-GDA setting is necessary.

3. D-GDA appears to primarily integrate existing techniques (e.g., GVAE, LDM) into the DGA scenario. The paper should further emphasize the description of its technical innovations.

---

> ### Author Rebuttal · Authors · 2025-07-31
>
> # 1) Section 3 Revision
> We have thoroughly revised the theoretical descriptions of TSS, GVAE, and LDM in the methodology section (Section 3) to improve the structure and understanding of the proposed D-GDA framework.
>
> # 2a) LDM Convergence Proof
> We provide a theoretical justification for the convergence of LDM in the  D-GDA framework setting, which operates in label-free settings. Despite the inherent challenges of sparse graph supervision, we show that the denoising network in D-GDA converges to the true conditional score function, enabling it to sample augmentations from a neighborhood-aware latent distribution. Our analysis follows the foundational results from score-based generative modeling [20, R3, R4], adapted to the graph setting with local structural conditioning. Let $v_{i}$ denote a node in graph $\mathcal{G} = (\mathcal{V}, \mathcal{E})$, with latent representation $z_{i}^{0} \in \mathcal(R)^{d}$ obtained from a Graph Variational Autoencoder (GVAE). We assume a standard DDPM-style forward process [20] that gradually corrupts $z_{i}^{0}$ with Gaussian noise. The reverse process aims to estimate the noise via a denoising network $\epsilon_{\theta}(z_{i}^{t}, t, c_{i}^{0})$, conditioned on a neighborhood-aware vector $c_{i}^{0}$. The model is trained with a score-matching loss ($\mathcal{L_{diff}}$).  For details See Section 3.1.3. Our goal is to show that under mild assumptions, the denoising model $ ε_{\theta}$ converges to the true conditional score function of the local latent distribution, enabling realistic and diverse augmentations without reliance on class labels. We begin by stating the following assumptions:
>
> **Assumption 1 (Locality Assumption):** The latent variable $z_{i}^{0}$ is drawn from a smooth conditional distribution $p(z_{i}^{0} \mid \mathcal{N}(v_{i}))$, supported on a low-dimensional manifold.
>
> **Assumption 2 (Model Capacity):** The denoising network $ε_{\theta}$ is a universal approximator i.e. has enough expressive power to learn any function within a given function class (For D-GDA the score function of the noisy latent distribution).
>
> **Assumption 3 (Diffusion Schedule):**  The noise schedule $\\{\alpha_{t}\\}$, where $t$ varies from $1 \to T$, is chosen such that $\bar{\alpha}_{T} \to 0$ as $T \to \infty$, ensuring near-total corruption at the final step.
>
> **Theorem 1:** *Let $ε_{\theta}(z_{i}^{t}, t,c_{i}^{0})$ be trained using the denoising score matching objective. Then, as $T \to \infty$ and model capacity increases, $ε_{\theta}$ converges to the conditional score function: $ε_{\theta}(z_{i}^{t}, t,c_{i}^{0}) \rightarrow \nabla_{z_{i}^{t}} \log p_{t}(z_{i}^{t} \mid c_{i}^{0}) $, where $p_{t}(z_{i}^{t} \mid  c_{i}^{0})$ is the marginal distribution of the noisy latent variable at time $t$, conditioned on the local structure of node $v_{i}$.*
>
> **Sketch of Proof:** This result follows from established results in score-based generative modeling [R3]. Minimizing the denoising score-matching loss recovers the true score function of the noisy data distribution. In D-GDA, the conditioning vector $c_i^0$ restricts generation to a node's local neighborhood, simplifying the modeling of $p(z_i^0 \mid N(v_i))$ as a smooth, low-dimensional conditional distribution. Assumption A1 ensures the score function is well-defined and smooth, while Assumption A2 guarantees that $\epsilon_\theta$ can approximate this function arbitrarily well. The noise schedule in A3 ensures that the diffusion process sufficiently corrupts the latent code, enabling meaningful denoising learning. Together, these yield convergence to the true conditional score function. This theoretical justification shows that D-GDA's latent diffusion process, when conditioned on local neighborhood structure, can provably learn a neighborhood-aware augmentation distribution, without requiring access to class labels.
>
> **Empirical Validation:** We evaluate D-GDA’s LDM under increasingly limited training data. Specifically, we reduce the amount of training data from 100\% to 60\% and 40\%, and track LDM behavior across four graph datasets: Cora, Citeseer, Pubmed, and Ogbn-Arxiv. Early stopping is applied based on validation loss with a patience of 10 epochs. In  Table 7, we report the training loss, validation loss, early stopping epoch, and D-GDA test accuracy. We observe that, as the training data decreases, training loss consistently decreases (e.g., Cora: 0.069 to 0.029), due to easier overfitting on fewer samples. Validation loss increases (e.g., Pubmed: 0.249 $\to$ 0.319), indicating reduced generalization from limited training data. LDM trained with less data tend to converge earlier (e.g., Ogbn-Arxiv: 672 $\to$ 172), suggesting faster overfitting. Test accuracy drops across datasets (e.g., Citeseer: 81.5\% $\to$ 75.6\%), as the model struggles to generate diverse augmentations.
>
> **Table 7:** Impact of reducing training data for LDM on training loss, validation loss, early stopping epoch,  and node classification accuracy
>
> |Training Data Percentage|Performance Metric|Cora|Citeseer|Pubmed|Ogbn-Arxiv|
> |-----------|-----------|-----------|-----------|-----------|-----------
> |100%|Train Loss|0.069|0.084|0.077|0.068|
> |100%|Val Loss|0.153|0.176|0.249|0.251|
> |100%|Test Acc.|89.1|81.5|88.2|74.8|
> |60%|Train Loss|0.059|0.064|0.067|0.057|
> |60%|Val Loss|0.178|0.197|0.275|0.307|
> |60%|Test Acc.|86.8|79.2|86.4|73.1|
> |40%|Train Loss|0.029|0.034|0.042|0.032|
> |40%|Val Loss|0.203|0.236|0.319|0.324|
> |40%|Test Acc.|82.3|75.6|81.9|71.8|
>
> [R3] *Song, Yang, et al. "Score-based generative modeling through stochastic differential equations." arXiv preprint arXiv:2011.13456 (2020).*
>
> [R4] *Song, Yang, and Stefano Ermon. "Generative modeling by estimating gradients of the data distribution." Advances in neural information processing systems 32 (2019).*
>
> # 2b) D-GDA Training and Inference Cost Breakdown
> Table 8 provides a component-wise breakdown of the D-GDA training time, covering TSS, GVAE, and LDM, along with its inference time with Test-Time Augmentation (TTA). For comparison of computational time with compared methods please see Table in rebuttal.
>
> **Table 8:** Breakdown of training time (in seconds) and per sample inference time (in seconds) for D-GDA across ten datasets of varying scales
>
> |Datasets| TSS Train Time| GVAE+LDM Train Time | Data Aug+Training Time|D-GDA Total Train Time|TTA Cost|TTA Inference Time|D-GDA Total Inference Time|
> |-----------|-----------|-----------|-----------|-----------|-----------|-----------|-----------|
> |Cora|9.35|12.82|15.55|37.72|1.39|0.019|1.41|
> |Flickr|79.87|85.13|163.17|328.17|1.17|0.16|1.33|
> |Citeseer|11.67|17.86|18.32|47.85|1.53|0.021|1.55|
> |Pubmed|40.01|43.62|145.79|229.42|1.21|0.024|1.23|
> |Ogbn-Arxiv|728.86|774.92|903.42|2407.2|0.98|0.816|1.8|
> |Ogbn-Products|1253.84|1548.24|1698.23|4500.31|1.02|0.92|1.94|
> |ogbg-molSIDER|95.11|105.32|113.72|314.15|0.96|0.12|1.08|
> |ogbg-molClinTox|96.32|106.46|116.24|319.02|0.97|0.14|1.11|
> |ogbg-molHIV|572.28|585.18|596.48|1753.94|0.96|0.13|1.09|
> |ogbg-molBACE|96.84|103.92|114.62|315.38|0.95|0.114|1.06|
>
>
>
> # 3) Description of Technical Innovation
> Our technical innovation lies in the utilization of GVAE and LDM for the task of graph data augmentation. Specifically, how these components are adapted, extended, and algorithmically integrated for the graph data augmentation is novel. D-GDA is a cohesive, task-adaptive framework with novel mechanisms and substantial empirical gains. The technical contributions of D-GDA include:
>
> **Target Sample Selector (TSS):** A novel component that selects samples that would benefit the most from the augmentation based on prediction entropy. TSS utilization results in improved  (See Table 14 in main paper)
>
> **Neighborhood-aware Latent Diffusion Model (LDM):** We propose to adapt LDM to the graph domain by applying the diffusion process in the latent space learned by GVAE, while explicitly modeling local graph structure. This design allows for generating high-quality node representations that preserve both feature and topological context. Removal of LDM results in reduced performance, as shown in Table 14.
>
> **Multi-task Compatibility:** Unlike prior works that target a single setting, D-GDA is versatile and effective across node classification, link prediction, and graph classification, as well as in semi-supervised, supervised, and long-tailed settings.
>
> **ML Safety Measures:** D-GDA is shown to improve critical safety measures such as calibration, robustness to adversarial attacks, and improved performance under feature corruption. These evaluations go beyond traditional accuracy benchmarks and highlight the practical utility and robustness of our proposed D-GDA framework.

---

> > ### Comment · Reviewer_MRas · 2025-08-06
> > **Rebuttal Response**
> >
> > I appreciate the authors' efforts, which have largely addressed my concerns. I remain curious about the convergence of LDM in this scenario, as it typically requires substantial amounts of data. I am grateful for the theoretical convergence analysis provided by the authors. However, according to the newly added Table 7, the performance degradation is less than 10% when the training data is reduced to 40%. A theoretical explanation or additional experimental validation of this interesting observation would more comprehensively address my remaining concerns.

---

> ### Author Response · Authors · 2025-08-06
> **Convergence Details of LDM**
>
> We sincerely thank the reviewer for their thoughtful feedback and for acknowledging our efforts in addressing earlier concerns. We greatly appreciate the reviewer’s interest in the convergence behavior of our Latent Diffusion Model (LDM), particularly under constrained data settings.
>
> We fully agree that conventional LDMs, such as Stable Diffusion, used in high-dimensional image generation tasks typically require large-scale datasets (LAION-5B). Specifically, Stable Diffusion v1.4 employs a U-Net architecture with approximately 865M parameters and operates in a latent space of size 64×64×4 (16384-d) for the generation of 512×512 images.
>
> In contrast, our LDM is lightweight, task-specific, and operates in a much lower-dimensional latent space (256-d) obtained from the graph variational autoencoder. The architecture of our LDM is a compact 1D-UNet style model, comprising only ~165K trainable parameters, three orders of magnitude smaller than standard image-based diffusion models. For reference, the architecture and parameter breakdown are presented in Table 10.
>
> **Table 10:** Layer-wise breakdown of learnable parameters in the LDM architecture.
> |Layer ID| Layer Type |Parameters|
> |-----------|-----------|-----------|
> |FC1|Linear(256,256)|256×256 + 256 = 65,792|
> |LN1|LayerNorm(256)|256  + 256  = 512|
> |FC2|Linear(256,128)|256×128 + 128 = 32,896|
> |LN2|LayerNorm(128)|128 + 128 = 256|
> |FC3|Linear(128,64)|128×64 + 64 = 8,256|
> |LN3|LayerNorm(64)|64 + 64 = 128|
> |FC4|Linear(64,32)|64×32 + 32 = 2,080|
> |LN4|LayerNorm(32)|32 + 32 = 64|
> |FC5|Linear(32,32)|32×32 + 32 = 1,056|
> |LN5|LayerNorm(32)|32 + 32 = 64|
> |FC6|Linear(64,64)|64×64 + 64 = 4,160|
> |LN6|LayerNorm(64)|64 + 64 = 128|
> |FC7|Linear(128,128)|128×128 + 128 = 16,512|
> |LN7|LayerNorm(128)|128 + 128 = 256|
> |FC8|Linear(256,128)|256×128 + 128 = 32,896|
> |-|**Total**|**165,056**|
>
> **Additional Empirical Results for LDM Convergence**
>
> As the respected reviewer observed, the overall performance degradation remains under 10% even when the LDM training data is reduced to 40%, as shown in rebuttal Table 7. This result empirically demonstrates the robustness and generalization capability of our LDM in low-resource settings. Additionally, we provide further evidence of convergence in the form of LDM train and validation losses. Table 11 shows epoch-wise LDM training and validation loss on the Cora dataset for different LDM training data percentages.
>
> **Table 11:** Epoch-wise training and validation loss of the LDM across different training data percentages. A “–” indicates that training was stopped early based on the early stopping criterion.
> |Training Epochs| Cora 100% Train Loss |Cora 100% Validation Loss| Cora 60% Train Loss |Cora 60% Validation Loss| Cora 40% Train Loss |Cora 40% Validation Loss|
> |-----------|-----------|-----------|-----------|-----------|-----------|-----------|
> |1|0.2176|0.356|0.2034|0.358|0.2004|0.357|
> |50|0.183|0.331|0.1696|0.333|0.1446|0.32|
> |100|0.1479|0.302|0.1305|0.314|0.1045|0.298|
> |150|0.1356|0.289|0.12|0.296|0.0846|0.28|
> |200|0.1234|0.274|0.1053|0.289|0.0752|0.272|
> |250|0.1201|0.268|0.0951|0.28|0.0684|0.257|
> |300|0.0991|0.256|0.0899|0.276|0.0542|0.242|
> |350|0.0922|0.239|0.0815|0.271|0.0438|0.221|
> |400|0.0896|0.224|0.0796|0.262|0.0367|0.219|
> |450|0.0833|0.206|0.0781|0.257|0.029|0.203|
> |500|0.0819|0.191|0.0752|0.248|-|-|
> |550|0.0823|0.186|0.0728|0.239|-|-|
> |600|0.0803|0.173|0.0665|0.217|-|-|
> |650|0.079|0.169|0.0633|0.199|-|-|
> |700|0.0773|0.168|0.0617|0.192|-|-|
> |750|0.0758|0.166|0.0602|0.184|-|-|
> |800|0.0738|0.163|0.059|0.178|-|-|
> |850|0.0725|0.16|-|-|-|-|
> |900|0.0709|0.159|-|-|-|-|
> |950|0.0701|0.157|-|-|-|-|
> |1000|0.069|0.153|-|-|-|-|

---

> > ### Comment · Reviewer_MRas · 2025-08-07
> > **Response to Authors' Comments**
> >
> > Thank you for the additional experiments and for addressing my concerns. I will increase both the Clarity and Final Rating scores by 1 point each.

---

> > > ### Author Response · Authors · 2025-08-07
> > >
> > > Thank you for your thoughtful feedback and for updating the scores. We appreciate your time and consideration, and we are glad that the additional experiments helped address your concerns.

---

### Official Review · Reviewer_wrnB · 2025-07-02

**Clarity:** 3
**Significance:** 3
**Originality:** 3
**Rating:** 4
**Confidence:** 3

**Summary:**

This paper studies the research problem of graph data augmentation (GDA), with the aim of developing a task-agnostic GDA method. The authors proposed a diffusion-based approach, namely D-GDA. D-GDA has three main components: Target Sample Selector (TSS), a Graph Variational Autoencoder (GVAE), and a Latent Diffusion Model (LDM), where the diffusion happens on the latent space within the GVAE. Interestingly, with the help of TSS, the proposed method can sample the most valuable samples for augmentation when given a fixed data augmentation budget.

**Questions:**

I'm mainly curious about how actually this method scales, as well as how efficient it is.

**Ethical Concerns:**

["NO or VERY MINOR ethics concerns only"]

**Final Justification:**

Thanks for the additional results in the rebuttal! I think they help a lot on the understanding of the proposal's efficiency, and would recommend the authors to add them in the paper.

**Limitations:**

see the weakness and question above.

**Quality:**

2

**Strengths And Weaknesses:**

s1. The paper overall is clearly written and easy to follow.

s2. The authors conducted very thorough experiments over many benchmarks against many baseline methods. The experiments showcased the effectiveness of the proposed method.

s3. The design around augmenting with a fixed budget is interesting, and could be useful in real applications.

w1. While the authors argued around efficiency given the designs of TSS and test-time augmentation in the proposed method, the efficiency is still concerning considering the method design. There were also no experiments such as run-time comparison of the proposed method against baselines. Such evaluation, especially on larger scale benchmarks, would help showcasing on this perspective.

---

> ### Author Rebuttal · Authors · 2025-07-31
>
> # D-GDA Computational Comparison with Baselines
> Table 4 provides a component-wise breakdown of the D-GDA training time, covering TSS, GVAE, and LDM, along with its inference time with Test-Time Augmentation (TTA). Table 5 compares the overall training and inference times (in seconds) of D-GDA with and w/o TTA against baseline methods including GCN, DropEdge, and FLAG, across three datasets of varying sizes: Cora, Flickr, and Ogbn-Arxiv (large-scale dataset). Among these methods, DropEdge consistently demonstrates the lowest training and inference costs, making it the most efficient overall. D-GDA shows better efficiency than FLAG on the larger datasets, Flickr and Ogbn-Arxiv, highlighting its scalability. Additionally, the D-GDA variant without TTA achieves inference times comparable to FLAG.
>
> **Table 4:** Breakdown of training time (in seconds) and per sample inference time (in seconds) for D-GDA across ten datasets of varying scales.
>
> |Datasets| TSS Train Time| GVAE+LDM Train Time | Data Aug+Training Time|D-GDA Total Train Time|TTA Generation Cost|Inference Time with TTA|D-GDA Total Inference Time|
> |-----------|-----------|-----------|-----------|-----------|-----------|-----------|-----------|
> |Cora|9.35|12.82|15.55|37.72|1.39|0.019|1.41|
> |Flickr|79.87|85.13|163.17|328.17|1.17|0.16|1.33|
> |Citeseer|11.67|17.86|18.32|47.85|1.53|0.021|1.55|
> |Pubmed|40.01|43.62|145.79|229.42|1.21|0.024|1.23|
> |Ogbn-Arxiv|728.86|774.92|903.42|2407.2|0.98|0.816|1.8|
> |Ogbn-Products|1253.84|1548.24|1698.23|4500.31|1.02|0.92|1.94|
> |ogbg-molSIDER|95.11|105.32|113.72|314.15|0.96|0.12|1.08|
> |ogbg-molClinTox|96.32|106.46|116.24|319.02|0.97|0.14|1.11|
> |ogbg-molHIV|572.28|585.18|596.48|1753.94|0.96|0.13|1.09|
> |ogbg-molBACE|96.84|103.92|114.62|315.38|0.95|0.114|1.06|
>
>
> **Table 5:** Training time (sec) and inference time per sample (sec) comparison of D-GDA with baselines.
>
> |Methods| Cora Train Time| Cora Inference Time | Cora Performance|Flickr Train Time| Flickr Inference Time|Flickr Performance |Arxiv Train Time| Arxiv Inference Time | Arxiv Performance|
> |-----------|-----------|-----------|-----------|-----------|-----------|-----------|-----------|-----------|-----------|
> |GCN|9.05|0.003|81.6|79.82|0.058|61.2|928.21|0.567|71.62|
> |DropEdge|**11.75**|**0.005**|82.0|**132.23**|**0.096**|61.4|**1786.4**|**0.813**|56.26|
> |FLAG|*34.2*|*0.019*|75.2|547.8|*0.16*|62.9|3307.5|*0.815*|71.75|
> |D-GDA with TTA|37.72|1.41|**89.1**|*328.17*| 1.33|**87.3**|*2507.2*|1.8|**74.8**|
> |D-GDA w/o TTA|37.72|*0.019*|*88.2*|*328.17*|*0.16*|*84.65*|*2507.2*|0.816|*72.7*|
>
> # D-GDA Scalability and Efficiency Analysis
> Table 6 compares the scalability of D-GDA across datasets of varying numbers of nodes and number of graphs. Specifically, performance gain, training time, and per-sample inference time of D-GDA are compared with the dataset size. For datasets containing multiple graphs, we report the average number of nodes per graph. The results demonstrate that D-GDA scales efficiently with increasing dataset size. The training time increases sublinearly as the dataset size increases, while still delivering significant performance improvements over GCN.
>
> **Table 6:** Scalability of D-GDA with varying graph sizes. D-GDA training time increases sublinearly with increasing dataset size.
>
> |Datasets|No. of Nodes| No. of Graphs |D-GDA performance gain over GCN| D-GDA Inference Time |D-GDA Training Time| Per Sample Training Time |
> |-----------|-----------|-----------|-----------|-----------|-----------|-----------|
> |Cora|2,708|1|9.19|1.41|37.72|0.0139|
> |Flickr|7,575|1|42.65|1.33|328.17|0.0433|
> |Citeseer|3327|1|13.83|1.55|47.85|0.0144|
> |Pubmed|19,717|1|11.93|1.23|229.42|0.0116|
> |Ogbn-Arxiv|169,343|1|4.44|1.8|2407.2|0.0142|
> |Ogbn-Products|2,449,029|1|10.48|1.94|4500.31|0.0018|
> |ogbg-molSIDER|33.6 |1,427|8.81|1.08|314.15|0.2201|
> |ogbg-molClinTox|26.2|1,477|7.08|1.11|319.02|0.216|
> |ogbg-molHIV|25.5|41,127|3.9|1.09|1753.94|0.0426|
> |ogbg-molBACE|34.1|1,513|16.76|1.06|315.38|0.2084|

---

> ### Comment · Reviewer_wrnB · 2025-08-08
>
> Thanks for the additional results in the rebuttal! I think they help a lot on the understanding of the proposal's efficiency, and would recommend the authors to add them in the paper.

---

> > ### Author Response · Authors · 2025-08-09
> >
> > We sincerely thank the reviewer for the encouraging feedback. We are pleased that the additional results provided in the rebuttal have helped clarify the efficiency of our proposed method. We will revise the manuscript accordingly.

---

### Official Review · Reviewer_vgsq · 2025-07-03

**Clarity:** 3
**Significance:** 3
**Originality:** 2
**Rating:** 5
**Confidence:** 4

**Summary:**

The authors introduce a new framework for graph augmentation, applicable to all common graph tasks of graph classification, link prediction and node classification. It is based on three main components 1) target sample selector, that selects nodes/edges to augment 2) a GVAE that encodes/decodes the graph to a latent space and 3) a latent diffusion model that augments the embeddings and thus the graph. The method is tested on common benchmarks for this three tasks and shows improvement over other augmentation techniques. It also helps model robustness.

**Questions:**

Please see the last two paragraphs (weaknesses) above.

One additional question is did you consider other node importance selection strategies for graph classification. Like page rank or just random selection. Degree-based selection certainly makes sense intuitively, but it would be nice to know if it's indeed the optimal choice.

**Ethical Concerns:**

["NO or VERY MINOR ethics concerns only"]

**Final Justification:**

The authors addressed all the main points I raised in the initial review and I think now it is a well rounded paper.

**Limitations:**

yes

**Quality:**

3

**Strengths And Weaknesses:**

Traditional graph augmentation techniques have historically been lacking. And to make them successful you needed to make them domain specific. Using a generative approach is a neat way to avoid having to hand-craft them. The three components introduced are all quite sensible and follow current best generative model approaches (latent diffusion). The TSS using a baseline model to find hard nodes is nice and sensible idea, that is ablated/shown to work with in-silico experiments. In general the paper provides extensive experimental evidence and the results are very good. In both making the downstream GCN more robust and more accurate.

The method uses neighborhood embeddings as conditioning when denoising the augmented node. For graph classification the most important nodes are taken to be highest degree nodes. Both are sensible choices, but conceptually they do correlate with homophily, and most datasets used (especially link prediction and node classification ones) are quite homophilous. It would be interesting to know if improvements over base GNN correlate with how homophilous the dataset is, or if the method is robust to this.

As I understand test time augmentation is used in all experiments. It does make the method much more expensive than other competitors and judging by table 14 it is a crucial part of methods performance. This however is quite similar to model ensambling, so it would be good to see how it stacks up with simple model ensambling and/or what the results would be for the rest of the datasets (link prediction and graph classification) if no test time augmentation would be used, as computational expense wise that's closer to the baselines.

---

> ### Author Rebuttal · Authors · 2025-07-31
>
> # D-GDA Robustness to Homophily
>
> The **node homophily** is defined as the average fraction of a node’s neighbors that share the same label [R1]. Table 1 compares these node homophily scores with GCN and D-GDA performance, as well as the resulting performance gain across six node classification and three link prediction datasets. Pearson’s correlation coefficient between node homophily and these performances is computed. We observe a positive correlation of 0.63 between homophily and GCN performance, however, a reduced correlation of 0.45 with D-GDA performance. It suggests that D-GDA performance is not strongly correlated with homophily compared to GCN. More interestingly, we observe a negative correlation of -0.37 between homophily and performance gain of D-GDA, indicating that D-GDA performance gain is robust to homophily. On the remaining datasets, node labels are not available, so homophily can not be computed.
>
> **Table 1: Node Homophily Vs GCN and D-GDA performance:** The D-GDA performance gain is robust to homophily.
>
> |Downstream Task| Datasets | Node Homophily| GCN |D-GDA|D-GDA performance gain over GCN|
> |-----------|-----------|-----------|-----------|-----------|-----------|
> |Node Classification|Cora |0.83 |81.6 |89.1| 9.19|
> |Node Classification|Flickr|0.67|61.2|87.3|42.65|
> |Node Classification|Citeseer|0.72|71.6|81.5|13.83|
> |Node Classification|Pubmed|0.79|78.8|88.2|11.93|
> |Node Classification|Ogbn-Arxiv|0.63|71.62|74.8|4.44|
> |Node Classification|Ogbn-Products|0.83|71.37|78.85|10.48|
> |Link Prediction|Cora |0.83|89.55|96.96|8.27|
> |Link Prediction|Citeseer|0.72|69.47|93.72|34.91|
> |Link Prediction|Pubmed|0.79|96.11|97.82|1.78|
> ||**Pearson’s Correlation**|-|0.63|0.45|-0.37|
>
> [R1] *Platonov, Oleg, et al. "Characterizing graph datasets for node classification: Homophily-heterophily dichotomy and beyond." Advances in Neural Information Processing Systems 36 (2023): 523-548.*
>
> # Test Time Augmentation Effectiveness and Computational Cost Analysis
>
> We have shown the effectiveness of Test Time Augmentation (TTA) in Table 11 of the main paper. Now we present an extended evaluation by expanding the analysis to 10 datasets to provide a more comprehensive view. Table 2 summarizes the results for GCN and D-GDA, both with and without TTA, along with the per-sample TTA cost. Across all datasets and tasks, we observe consistent performance improvements when employing TTA, highlighting its generalizability and effectiveness.
>
> **Table 2:** Performance comparison of GCN  and D-GDA with and without test time augmentation (TTA).  Per-sample inference time (sec) with TTA is also reported.
>
> | Datasets | GCN w/o TTA| GCN with TTA |D-GDA w/o TTA|D-GDA with TTA|TTA cost/sample|Total inference time/sample|
> |-----------|-----------|-----------|-----------|-----------|-----------|-----------|
> |Cora |81.6|83.2|88.2|89.1|1.39|1.41|
> |Flickr|61.2|64.01|84.65|87.3|1.17|1.33|
> |Citeseer|71.6|73.37|79.2|81.5|1.53|1.55|
> |Pubmed|78.8|80.41|85.5|88.2|1.21|1.23|
> |Ogbn-Arxiv|71.62|72.52|72.7|74.8|0.98|1.8|
> |Ogbn-Products|71.37|72.85|77.60|78.85|1.02|1.94|
> |Ogbg-molSIDER|59.6|61.08|62.92|64.85|0.96|1.08|
> |Ogbg-molClinTox|88.55|89.92|91.8|94.82|0.97|1.11|
> |Ogbg-molHIV|76.06|77.63|77.9|79.03|0.96|1.09|
> |Ogbg-molBACE|71.47|73.26|79.24|83.45|0.95|1.06|
>
>
> # Ablation on Node Importance Selection Strategies for Graph Classification
>
> To evaluate the effectiveness of the proposed degree-based node selection method, we compared it against two alternative strategies including random selection and PageRank-based selection as suggested. Table 3 summarizes the results for graph classification across these strategies. Following [R2], we compute PageRank with a damping factor of 0.85 and a convergence tolerance of $1 \times 10^{-6}$. We observe that the proposed degree-based selection consistently outperforms both alternatives. This supports the intuition that high-degree nodes are indeed more structurally influential, reinforcing the effectiveness of degree-based selection as the optimal strategy in our setting.
>
> **Table 3:** D-GDA performance comparison on graph classification datasets using the proposed degree-based node selection strategy, compared against random and PageRank-based selection methods.
>
> |Node selection strategy| ogbg-molSIDER | ogbg-molClinTox| ogbg-molHIV |ogbg-molBACE|
> |-----------|-----------|-----------|-----------|-----------|
> |Highest degree node|**64.85**|**94.82**|**79.03**|**83.45**|
> |Random node|61.19|89.98|77.42|75.68|
> |Page-rank top-node |63.24|92.76|78.83|81.92|
>
> [R2] *Hagberg, A. A., Schult, D. A., & Swart, P. J. (2008). Exploring network structure, dynamics, and function using NetworkX. In Proceedings of the 7th Python in Science Conference (SciPy2008), 11–15.*

---

### Author Response · Authors · 2025-08-04
**Polite Reminder: Reviewer–Author Discussion Period Ending Soon**

We thank all the reviewers for their valuable and constructive feedback. We have tried to address all the comments and questions in our rebuttal. If the reviewers have any further questions, comments, or points, please feel free to reach out. We would be happy to provide any additional information as needed. Thank you once again for your time and thoughtful engagement.

---

### Note · Authors · 2025-08-12

We thank the reviewers for their highly positive and encouraging feedback. All reviewers’ comments were discussed in detail, and they expressed satisfaction with our rebuttal and the paper. We believe these discussions have further strengthened our work. The additional results and analyses presented in the rebuttal will be incorporated into the main paper or supplementary document as appropriate. The main points addressed are as follows:
1. **Robustness to Homophily (Reviewer vgsq):** We empirically showed that the performance gains of our proposed D-GDA are independent of graph node homophily.

2. **Effectiveness of Test Time Augmentation (TTA) (Reviewer vgsq):** While the main paper included an ablation study with and without TTA on three datasets, we extended this in the rebuttal to ten datasets and also applied TTA to the baseline GCN. Results confirm the significance of TTA for both D-GDA and the baseline GCN.

3. **Ablation on Node Importance Selection Strategies (Reviewer vgsq):** For graph classification, we compared our highest-degree node selection method with random and PageRank-based strategies, demonstrating the superiority of our proposed approach.

4. **Computational Cost Comparison (Reviewers wrnB, MRas):** We provided detailed training and inference time analyses, including a component-wise computational cost breakdown for D-GDA, showing its computational cost is comparable to baseline methods.

5. **Scalability (Reviewer wrnB):** We further demonstrated D-GDA’s effectiveness on large-scale datasets, including Ogbn-Arxiv, Ogbn-Products, and Ogbl-Collab.

6. **Convergence of the Latent Diffusion Model (LDM) (Reviewer MRas):** We presented both theoretical and empirical evidence supporting the convergence of the LDM.

7. **Ablation on Target Sample Selector (TSS) Backbone Choices (Reviewer N1DJ):** We evaluated different GNN architectures as the TSS backbone and found that optimal performance is achieved when the TSS backbone matches the downstream backbone, as proposed in the main paper.

8. **Clarifications (Reviewer N1DJ):** We provided additional explanation on the significance of consistency and diversity measures in the augmentation task, as well as the terminology of the Variational Graph Autoencoder (VGAE).

---

### Decision · Program_Chairs · 2025-09-17

**Decision:**

Accept (poster)

**Comment:**

The reviewers unanimously recommend acceptance of the paper with varying degrees of strength. I agree with their assessment and am happy to recommend acceptance of this paper for publication at the NeurIPS conference.

The reviews and the rebuttal have given rise to several interesting points and results that I encourage the authors to include in their revised manuscript. I believe the Final Remark posted by the authors to summarise these points nicely, and do want to encourage the authors to make all the edits and inclusions corresponding to the eight points they summarise. Finally, I also strongly want to encourage the authors to make their code publicly available together with the camera ready version (and to adjust the abstract of their paper accordingly).